statistics/health and disease and epidemiology/differential equations

Bayesian inference, Markov chain Monte Carlo, approximate Bayesian computation, compartmental models, ordinary differential equations, epidemiology models

**Author for correspondence:**
Amani A. Alahmadi
e-mail: amani.alahmadi@monash.edu

# A comparison of approximate versus exact techniques for Bayesian parameter inference in nonlinear ordinary differential equation models

Amani A. Alahmadi[1,4], Jennifer A. Flegg[2], Davis G. Cochrane[1], Christopher C. Drovandi[3] and Jonathan M. Keith[1]

[1]School of Mathematics, Monash University, Clayton, Victoria, Australia
[2]School of Mathematics and Statistics, University of Melbourne, Parkville, Victoria, Australia
[3]School of Mathematical Sciences, Queensland University of Technology, Brisbane, Queensland, Australia
[4]College of Science and Humanities, Shaqra University, Shaqra, Saudi Arabia

AAA, 0000-0001-5359-1340; JAF, 0000-0002-8809-726X; CCD, 0000-0001-9222-8763; JMK, 0000-0002-9675-3976

The behaviour of many processes in science and engineering can be accurately described by dynamical system models consisting of a set of ordinary differential equations (ODEs). Often these models have several unknown parameters that are difficult to estimate from experimental data, in which case Bayesian inference can be a useful tool. In principle, exact Bayesian inference using Markov chain Monte Carlo (MCMC) techniques is possible; however, in practice, such methods may suffer from slow convergence and poor mixing. To address this problem, several approaches based on approximate Bayesian computation (ABC) have been introduced, including Markov chain Monte Carlo ABC (MCMC ABC) and sequential Monte Carlo ABC (SMC ABC). While the system of ODEs describes the underlying process that generates the data, the observed measurements invariably include errors. In this paper, we argue that several popular ABC approaches fail to adequately model these errors because the acceptance probability depends on the choice of the discrepancy function and the tolerance without any consideration of the error term. We observe that the so-called posterior distributions derived from such methods do not accurately reflect the epistemic uncertainties in parameter values. Moreover, we demonstrate that these methods provide minimal computational advantages

over exact Bayesian methods when applied to two ODE epidemiological models with simulated data and one with real data concerning malaria transmission in Afghanistan.

## 1. Introduction

Models of dynamical systems consisting of sets of ordinary differential equations (ODEs) are an essential tool to describe many processes in science and engineering. ODE models contain parameters such as kinetic rates and initial concentrations. However, these parameters often cannot be measured directly by experiments, or there is inherent uncertainty in the parameter values. As such, these parameter values need to be estimated using statistical techniques such as maximum likelihood estimation or Bayesian inference. In the last decade much research has focused on estimating the unknown parameters of ODE systems under a Bayesian framework. One reason is that the Bayesian approach provides appropriate quantification of the uncertainty of parameters (and hence model predictions) through the posterior distribution.

Markov chain Monte Carlo (MCMC) techniques were first developed by Metropolis *et al.* [1]. This class of technique creates a Markov chain which has the posterior distribution as its limiting distribution. The state of the chain after a number of steps is used as a sample from the posterior distribution and the quality of this sample improves as the number of steps gets larger. The original algorithm proposed by Metropolis was generalized by Hastings [2] to give the Metropolis–Hastings algorithm.

Exact Bayesian inference techniques have grown steadily more sophisticated over time, increasing the efficiency and complexity of sampling schemes. The modern Bayesian toolbox now includes schemes such as sequential Monte Carlo (SMC) [3], the Metropolis adjusted Langevin algorithm (MALA) [4] and hybrid (Hamiltonian) Monte-Carlo (HMC) [5,6]. These schemes improve the Metropolis–Hastings algorithm, enabling efficient sampling from high dimensional, strongly correlated posterior distributions.

However, there are many models that possess a computationally intractable likelihood function, ruling out exact Bayesian methods. This has led to the development of approximate Bayesian computation (ABC). The ABC methodology first appeared as the ABC rejection algorithm [7] which avoids calculation of the likelihood function. The theory was generalized and substantiated by Beaumont *et al.* [8]. To obtain samples more efficiently, a MCMC approach to ABC was formulated by Marjoram *et al.* [9].

In the context of dynamical systems, both approximate and exact Bayesian techniques involve numerical solution of the set of ODEs for each proposed set of parameters in order to evaluate how well the numerical solution matches the observed data. A desire to avoid the computational costs associated with numerical solution of the ODEs has led to the development of Gaussian Process (GP) models [10–12] for ODE parameter inference.

Dass *et al.* [13] proposed a two-step method to approximate the posterior distribution of unknown parameters in an ODE model. In the first step, data are generated from the ODE using a numerical method and then the second step uses the Laplace approximation to marginalize the posterior for each parameter. This method gives a fast approach compared to a full Bayesian computational scheme.

ABC methods based on SMC have been proposed [14] and many authors have developed approaches to improve the performance of the SMC ABC algorithm (see for example Beaumont *et al.* [15]). An SMC ABC approach was developed by Toni *et al.* [16], with application to dynamical systems. Their algorithm is theoretically sound, but we question the validity of the Bayesian posteriors they produce when they apply ABC to several examples involving ODE models. The authors apply ABC where they take the observed data as synthetically generated, where the ODE model is solved at an assumed true parameter value and measurement error added. However, when 'simulating' data in their ABC procedure, the ODE model is solved only, without generating measurement error. In this paper, we show that such an approach generates parameter distributions that are sensitive to the ABC tolerance, and will eventually converge onto a point mass if the tolerance is continually reduced. Thus this approach fails to correctly characterize the uncertainty as a Bayesian approach would aim to do.

In order to 'correctly' apply ABC to ODE models, one must simulate from the assumed measurement error model after solving the ODE. However, we also show in this paper that an exact Bayesian approach is more computationally efficient than this 'correct' ABC implementation, questioning the need for considering ABC in the first place when attempting to estimate the posterior distribution for ODE models.

Given that the Toni *et al.* [16] paper is highly cited, we are concerned that other researchers might follow their ABC approach for calibrating ODE models. For example, Gupta *et al.* [17] compared the performance of MCMC, parallel tempering (PT) and SMC ABC (using the ABC Sys-Bio package) in

estimating parameters in ODE models. The authors analysed simulated data with measurement error added, taken from the ABC-SysBio package. Then when applying MCMC and PT to infer model parameters, they assumed 'a likelihood function with 1% Gaussian error'. However, when using SMC ABC, no noise was added to the simulated data. Consequently, not only does this make the comparison invalid, but also the resulting approximate posterior distribution produced by SMC ABC does not represent the uncertainty around the parameter values. In another example, Silk *et al.* [18] present applications to molecular dynamical systems in which they 'have focused on the sequential ABC algorithm proposed by Toni *et al.* [16]'. Silk *et al.* [18] mention that they simulate the model 'subject to some small added zero-mean Gaussian noise with covariance 0.01*I*' so they have clearly used 'noise added in the simulation step, $\sigma$ is considered known' (option 2 from Toni's thesis [19, p. 154]). However, for many real problems, this is not practical and we fear that ABC users might revert to the option of simulating without noise. Two other examples of assuming known noise are in da Costa *et al.* [20] and Costa *et al.* [21]. The authors assumed the uncertainties to be 'additive, uncorrelated, Gaussian, with zero mean and' a known standard deviation, as they stated on pages 2801 and 1295, respectively. Moreover, Barnes *et al.* [22] presented an implementation of ABC SMC for ODEs (section 4.1 of their paper) and used the SysBio package. We show in our paper that using this package with ODE models can give an incorrect approximation to the posterior when not considering estimation of the noise. There is no explanation in Barnes *et al.* [22] regarding the authors' assumption about the noise. The same issue appeared in Toni & Stumpf [23] and Sun *et al.* [24]: they applied SMC ABC for an ODE model, but there are no details regarding the authors' assumptions about the noise. Understanding the overall noise (uncertainty) associated with the unknown parameter values when conducting parameter estimation using ODE models is important, especially when we aim to use these ODE models to inform real-world applications.

The remainder of this paper is organized as follows. In §2, we introduce a simple method of exact Bayesian inference and two methods of approximate Bayesian computation (MCMC ABC and SMC ABC), complemented with a discussion on the approximation to a point mass that results from SMC ABC and MCMC ABC. Application of MCMC, MCMC ABC and SMC ABC to two ODE epidemiological models with simulated data and one with real epidemiological data are presented in §3. Section 4 presents further discussion, comparison of the presented methods and our conclusions.

# 2. Bayesian techniques for ODE parameter inference

Bayesian techniques such as Markov chain Monte Carlo (MCMC) methodologies are sampling-based methods that involves sampling the posterior density

$$p(\boldsymbol{\theta}|\mathbf{y}) \propto p(\mathbf{y}|\boldsymbol{\theta})p(\boldsymbol{\theta}), \tag{2.1}$$

or an approximation to equation (2.1) in the case of approximate Bayesian computation (ABC) approaches, to calculate the desired density, where $\mathbf{y} = (y_1, \ldots, y_n)$ is the observed data, $\boldsymbol{\theta}$ are the unknown parameters, $p(\boldsymbol{\theta}|\mathbf{y})$ is the posterior distribution, $p(\mathbf{y}|\boldsymbol{\theta})$ is the likelihood and $p(\boldsymbol{\theta})$ is the prior. In this section we discuss application of these Bayesian frameworks in the context of inferring parameters for ODE models.

MCMC techniques as developed by Metropolis *et al.* [1] and Hastings [2] can be used to sample from the posterior distribution in equation (2.1). The Metropolis–Hastings algorithm constructs a Markov chain for which the stationary and limiting distribution is the posterior distribution. After running the chain for a sufficient amount of time,[1] samples from the chain can be considered draws from the posterior distribution. An implementation of the Metropolis–Hastings algorithm is given in appendix A. However, MCMC methods require the computation of the likelihood function, $p(\mathbf{y}|\boldsymbol{\theta})$, in equation (2.1). As a result, ABC methods were developed to sample from an approximation to the posterior in cases for which the likelihood is intractable or too computationally costly to compute. Instead of calculating the likelihood as before, a distance between the observed data, $\mathbf{y}$, and simulated data, $\mathbf{z}$, is calculated and for sufficiently small distance the parameter proposals are accepted. For more explanation see appendix A. ABC targets an approximate posterior [26]:

$$p_\epsilon(\boldsymbol{\theta}, \mathbf{z}|\mathbf{y}) \propto \mathbb{1}(\rho(\mathbf{z}, \mathbf{y}) \leq \epsilon)p(\boldsymbol{\theta})f(\mathbf{z}|\boldsymbol{\theta}), \tag{2.2}$$

---

[1]Sufficient time in the context of MCMC can be taken to mean that the chain is close to convergence. In practice this is often assessed by checking that multiple chains produce a Gelman–Rubin diagnostic less than 1.05 [25].

where $\mathbb{1}$ is an indicator function that takes the value one if its logical argument is true and zero otherwise and $f(\mathbf{z} \mid \boldsymbol{\theta})$ is the model that generates simulations $\mathbf{z}$ giving $\boldsymbol{\theta}$. The accuracy of ABC approaches depends on choosing a suitable discrepancy function $\rho(\mathbf{z}, \mathbf{y})$ and an appropriate tolerance $\epsilon$ [9]. In practice, the discrepancy function typically compares sets of summary statistics $s(\cdot)$ for the observed and simulated datasets. ABC rejection sampling is very simple to implement, though it can suffer from extremely low acceptance rates when the prior distribution is dissimilar to the posterior distribution [9]. To counteract this deficiency, a more efficient ABC technique based on MCMC was developed [9]. For more details see appendix A. Furthermore, in order to improve the low acceptance rate in the basic ABC algorithm, an SMC ABC algorithm was proposed in Sisson *et al.* [14], based on the SMC sampler methodology developed by Del Moral *et al.* [27]. The SMC ABC algorithm converges to the approximate posterior distribution through a number of intermediate distributions with a distance threshold that is sequentially decreased, see appendix A. The efficiency of the SMC ABC algorithm depends not only on the model complexity and the amount of data available, but also on the choice of the decreasing sequence of $\epsilon_t$ (the tolerances), and the choice of perturbation kernel $K_t$, according to Filippi *et al.* [28]. There are various ways to construct the decreasing sequence of $\epsilon_t$, either manually or adaptively as proposed in Drovandi & Pettitt [29] and Del Moral *et al.* [30]. In the adaptive method, the value of $\epsilon_t$ is chosen to be the $\alpha$th quantile of the discrepancies between the observed data and the simulated data that was generated in the $(t-1)$th population (see appendix A), where $0 \le \alpha \le 1$. In this paper, we used the latter method of selecting the sequence of tolerance thresholds and we stopped the algorithm when we reached a final $\epsilon_t$ that setting the desired final agreement between simulated and real data Liepe *et al.* [31].

The choice of perturbation kernel affects the acceptance rate in SMC ABC and the time consumed by the algorithm as explained in Filippi *et al.* [28]. Perturbation kernels can be divided into two classes: component-wise perturbation kernels and multivariate perturbation kernels. For component-wise perturbation kernels, one can use a uniform distribution or a univariate Gaussian distribution to perturb the particle $\boldsymbol{\theta}^*$ sampled from the previous population $\{\boldsymbol{\theta}_{t-1}^{(i)}\}_{i=1}^{N}$. The standard deviation of the kernel can be fixed in advance for each population, but more recently practitioners are adaptively choosing the width of the kernel (Beaumont *et al.* [15], Didelot *et al.* [32], Filippi *et al.* [28]).

If the model parameters are correlated, a component-wise perturbation kernel can fail to capture the structure of the true posterior, leading to a low acceptance rate. To overcome this problem, a multivariate normal distribution with a covariance matrix $\boldsymbol{\Sigma}^{(t)}$ that depends on the covariance of the previous populations can be used to perturb the particles [28]:

$$\boldsymbol{\Sigma}^{(t)} = \sum_{i=1}^{N} \sum_{k=1}^{N_0} w_{t-1}^{(i)} \hat{w}^{(k)} (\hat{\boldsymbol{\theta}}^{(k)} - \boldsymbol{\theta}_{t-1}^{(i)})(\hat{\boldsymbol{\theta}}^{(k)} - \boldsymbol{\theta}_{t-1}^{(i)})^T, \tag{2.3}$$

where $\{\hat{\boldsymbol{\theta}}^{(k)}\}_{1 \le k \le N_0}$ are the particles from the previous populations for which the corresponding simulated data $z^{(k)}$ satisfy $\rho(z^{(k)}, y) < \epsilon_t$ (remembering $\epsilon_t < \epsilon_{t-1}$) and $\hat{w}^{(k)}$ are the associated weights.

To further improve the performance of SMC ABC, we adopted a method proposed in Prangle [33] to adaptively update the discrepancy function, $\rho(\mathbf{z}, \mathbf{y})$. We used a weighted Euclidean distance function:

$$\rho(\mathbf{z}, \mathbf{y}) = \sum_{j=1}^{n} \left( \frac{z_j - y_j}{\zeta_j} \right)^2, \tag{2.4}$$

where $y_j$ is the $j$th observation, $z_j$ is the $j$th simulated observation in the simulated data $z = (z_1, \ldots, z_n)$ and $\zeta_j$ is a tunable scaling factor that allows the contribution to the discrepancy function of the $j$th coordinate to be normalized. The reason for normalizing the coordinates is to prevent any of them dominating the acceptance decision in the algorithm. In non-adaptive methods, the values of $\zeta_j$ are determined in advance and fixed. Fixing $\zeta_j$ from the first iteration in SMC ABC will not guarantee that the $j$th coordinate will be normalized in later iterations because in SMC ABC after the first round we are not sampling from the prior so the scale to normalize needs to be adapted.

To adapt the values of $\zeta_j$ in each iteration, Prangle [33] proposed calculating the median absolute deviation (MAD) of the $j$th coordinate of the simulated data vectors from the previous iteration (including those rejected). The value of the next $\epsilon_t$ is also determined using these distances; for more details see algorithm 4 in [33]. Note that Prangle [33] defined the discrepancy function in terms of summary statistics for $z$ and $y$, as is usual in ABC. Here we have used the coordinates of $z$ and $y$ directly, following the approach of Toni *et al.* [16] for inference of ODE model parameters.

Vaart *et al.* [34] proposed an ABC method called *error-calibrated ABC* that implements a general methodology introduced by Wilkinson [35]. In their method, they incorporated the estimation of the noise into the ABC technique by identifying an ABC acceptance probability in which the noise is assumed to be normally distributed and independent. This results in improved estimates of the parameter values and their uncertainty. An implementation of the error-calibrated ABC algorithm is given in algorithm 1.

---

**Algorithm 1.** Error-calibrated ABC algorithm, Vaart *et al.* [34].

---

1: Repeat *n* times:.

   (a) Draw $\theta^* \sim p(\theta)$.

   (b) Simulate $\mathbf{z}^*$ from model given $\theta^*$.

2: Find $\hat{\mathbf{z}}$, the simulated value that minimizes $\rho(\mathbf{z}, \mathbf{y})$.

3: For each data type $k$, calculate $\hat{\lambda}_k$, the standard deviation of all corresponding $\hat{\mathbf{z}}_n - \mathbf{y}$.

4: Accept $(\theta^*, \mathbf{z}^*)$ with probability $p_{\chi_I^2}(\mathbf{s})\mathbf{s}^{1-\frac{1}{2}}/\mathbf{c}$, where $\mathbf{s} = \sum_{j=1}^{I}\left(\dfrac{\mathbf{z}_j^* - \mathbf{y}_j}{\hat{\lambda}_k}\right)^2$ and $\mathbf{c}$ is equal to the maximum acceptance probability across all runs.

---

## 2.1. Bayesian inference for parameters in ODE models

Consider a $Q$-dimensional dynamical system for the state variable vector, $\mathbf{x}(t)$, described by the system of ODEs:

$$\dot{\mathbf{x}}(t) = \mathbf{f}(\mathbf{x}, \boldsymbol{\theta}, t), \tag{2.5}$$

where $\mathbf{x}$ is a $Q \times 1$ vector of the dependent variables, $\mathbf{f}$ is a $Q \times 1$ vector-valued Lipschitz continuous function with respect to $\mathbf{x}$, $\boldsymbol{\theta}$ is an $M \times 1$ vector of model parameters, $t$ is the independent variable (often time) and $\dot{\mathbf{x}}$ represents the derivative of $\mathbf{x}$ with respect to the independent variable. Given the dynamical system in equation (2.5), along with values for the parameter vector, $\boldsymbol{\theta}$, and the initial condition, $\mathbf{x}_0$, the solution to the system can be approximated numerically.

We denote an experimental observation at time $t_k$ by the $Q \times 1$ vector $\mathbf{y}_k$. Experimental observations are taken at $K$ time points; the times are stored in a $K \times 1$ vector $\mathbf{t} = (t_1, t_2, \ldots, t_K)^T$ and the observations are stored in the $Q \times K$ matrix $\mathbf{y} = (\mathbf{y}_1, \mathbf{y}_2, \ldots, \mathbf{y}_K)$. These observations are usually associated with some unknown noise process, characterized by one or more variance parameters, say $\boldsymbol{\sigma}^2$. The (approximate) solution for the dependent variables at time $t_k$, given $\boldsymbol{\theta}$ and $\mathbf{x}_0$, is denoted by the $Q \times 1$ vector $\hat{\mathbf{x}}(t_k; \boldsymbol{\theta}, \mathbf{x}_0)$. The solution for the dependent variables at times $\mathbf{t}$ is stored in the $Q \times K$ matrix $\hat{\mathbf{x}}(\mathbf{t}; \boldsymbol{\theta}, \mathbf{x}_0) = (\hat{\mathbf{x}}(t_1; \boldsymbol{\theta}, \mathbf{x}_0), \hat{\mathbf{x}}(t_2; \boldsymbol{\theta}, \mathbf{x}_0), \ldots, \hat{\mathbf{x}}(t_K; \boldsymbol{\theta}, \mathbf{x}_0))$. In a Bayesian setting, the posterior distribution for $\boldsymbol{\theta}$ and $\boldsymbol{\sigma}^2$ given $\mathbf{y}$ is:

$$p(\boldsymbol{\theta}, \boldsymbol{\sigma}^2|\mathbf{y}) \propto p(\mathbf{y}|\boldsymbol{\theta}, \boldsymbol{\sigma}^2)p(\boldsymbol{\theta})p(\boldsymbol{\sigma}^2), \tag{2.6}$$

where $p(\mathbf{y}|\boldsymbol{\theta}, \boldsymbol{\sigma}^2)$ is the likelihood, $p(\boldsymbol{\theta})$ and $p(\boldsymbol{\sigma}^2)$ are independent priors for $\boldsymbol{\theta}$ and $\boldsymbol{\sigma}^2$ respectively.

### 2.1.1. Observation model

In this paper, we assume that each observation, $\mathbf{y}_k$ for $k = 1, \ldots, K$, has an associated additive noise process, $\boldsymbol{\delta}_k$, such that

$$\mathbf{y}_k = \hat{\mathbf{x}}(t_k; \boldsymbol{\theta}, \mathbf{x}_0) + \boldsymbol{\delta}_k, \tag{2.7}$$

where $\boldsymbol{\delta}_k$ is a $Q \times 1$ vector and $\hat{\mathbf{x}}(t_k; \boldsymbol{\theta}, \mathbf{x}_0)$ is the solution for the dependent variables at time $t_k$, given $\boldsymbol{\theta}$ and $\mathbf{x}_0$. Under a Gaussian error model (we assumed Gaussian model for simplicity and illustration purposes but we can assume any kind of error model), and assuming the $\boldsymbol{\delta}_k$ are independent of each other, $\mathbf{y}_k$ follows a multivariate normal distribution:

$$\mathbf{y}_k \sim \text{MVN}(\hat{\mathbf{x}}(t_k; \boldsymbol{\theta}, \mathbf{x}_0), \boldsymbol{\Sigma}(\boldsymbol{\sigma}^2)), \tag{2.8}$$

where $\boldsymbol{\Sigma}(\boldsymbol{\sigma}^2)$ is a diagonal matrix with diagonal elements $\boldsymbol{\sigma}^2 = (\sigma_1^2, \sigma_2^2, \ldots, \sigma_Q^2)^T$ associated with the $Q$ dependent variables. Hence, the likelihood function is given by

$$\mathcal{L}(\mathbf{y}|\hat{\mathbf{x}}(\mathbf{t}; \boldsymbol{\theta}, \mathbf{x}_0), \boldsymbol{\sigma}^2) = \prod_{k=1}^{K} \text{MVN}(\mathbf{Y}_k; \hat{\mathbf{x}}(t_k; \boldsymbol{\theta}, \mathbf{x}_0), \boldsymbol{\Sigma}(\boldsymbol{\sigma}^2)) \tag{2.9}$$

and the posterior density is

$$p(\boldsymbol{\theta}, \boldsymbol{\sigma}^2, \mathbf{x}_0|\mathbf{y}) \propto p(\boldsymbol{\theta})p(\boldsymbol{\sigma}^2)p(\mathbf{x}_0) \prod_{k=1}^{K} \text{MVN}(\mathbf{Y}_k; \hat{\mathbf{x}}(t_k; \boldsymbol{\theta}, \mathbf{x}_0), \boldsymbol{\Sigma}(\boldsymbol{\sigma}^2)). \tag{2.10}$$

The Bayesian techniques discussed in appendix A can be used to sample from the posterior distribution of $\{\boldsymbol{\theta}, \mathbf{x}_0, \boldsymbol{\sigma}^2\}$ in the case of MCMC and an approximation to the posterior in the case of ABC methods.

## 2.2. Model misspecification in ABC methods for ODE models

Exact Bayesian methods, such as MCMC, generate samples directly from $p(\boldsymbol{\theta}, \boldsymbol{\sigma}^2|\mathbf{y})$ (at least in the limiting sense), rather than from the approximate posterior $p_\epsilon(\boldsymbol{\theta}, z|y)$ shown in equation (2.2). This applies in general, but in particular for the case of parameter inference in ODE models, as long as an observational model has been defined, leading to a posterior distribution such as in equation (2.10). Evaluating this expression requires solving a system of ODEs to obtain $\hat{\mathbf{x}}(t_k; \boldsymbol{\theta}, \mathbf{x}_0)$ for a specific collection of parameters. It is then straightforward, at least for this simple noise model, to evaluate the likelihood and the posterior density up to a normalizing constant.

One of the major motivations for using likelihood-free methods, such as ABC, is that they are applicable even when evaluating the likelihood is difficult or impossible. That motivation is not present here, since solving the system of ODEs for each proposed parameter vector $\boldsymbol{\theta}$ is the main computational burden involved in evaluating equation (2.10), and this is still necessary for ABC, at least using the method proposed by Toni *et al.* [16]. Their method does still have a computational advantage, in that it avoids evaluating the density of the noise model, which may be prohibitive for certain models. However, where the simple independent Gaussian noise model of equation (2.2) is appropriate, or some other simple noise model applies, the contribution of these density evaluations to the overall computational burden will be negligible.

The method of Toni *et al.* [16] actually avoids even simulating draws from the noise model. This is made clear in the following text which appears in the thesis by Toni [19, p. 154], which is the basis of the work in Toni *et al.* [16]: 'We explore the differences between three different inference approaches:

1. No noise added in the simulation step, $\theta$ is the unknown parameter. This framework has been introduced in Chapter 3 and used throughout this thesis.
2. Noise added in the simulation step, $\sigma$ is considered known and $\theta$ unknown.
3. Noise added in the simulation step, both $\sigma$ and $\theta$ are unknown.'

Since option 1 (the approach taken throughout the thesis and associated papers) avoids adding noise in the simulation step, the method is applicable regardless of the noise model. However, this generality comes at a cost and, as we explain in this section, results in an approximate distribution of the form equation (2.2) that contains no information about parameter uncertainty.

Toni *et al.* [16] adapted the Sisson *et al.* [14] SMC ABC algorithm and used it to infer parameters in ODE models. However, the method they devised differs in two crucial aspects from standard practice in implementing ABC. The first is that they do not simulate data vectors $\mathbf{z}^*$ from the same model they assume for the data, which is of the form shown in equation (2.9). Instead, they generate $\mathbf{z}^*$ by merely solving the underlying system of ODEs for each proposed value of the parameter vector $\boldsymbol{\theta}^{**}$. The simulated data $\mathbf{z}^*$ is thus a deterministic function of $\boldsymbol{\theta}^{**}$, without any added noise, and in effect the underlying likelihood distribution model used in the resulting ABC algorithm is a point mass concentrated at the solution of the system of ODEs. In this sense, the data generation model used by Toni *et al.* [16] is misspecified.

A second departure from standard ABC practice is that the discrepancy function used by Toni *et al.* [16] directly computes a distance between the simulated and observed data, originally using Euclidean distance

$$\rho(\mathbf{z}, \mathbf{y}) = \sum_{j=1}^{n} (z_j - y_j)^2, \tag{2.11}$$

but they also experimented with alternative metrics. In this paper, we experiment with the more general adaptively weighted distance function in equation (2.4). All of these discrepancy functions have in

common that they can only take a zero value when the simulated data $\mathbf{z}^*$ exactly corresponds to the observed data $\mathbf{y}$. In contrast, practical ABC methods more commonly use a discrepancy function based on the distance between vectors of summary statistics $s(\mathbf{z}^*)$ and $s(\mathbf{y})$, which have much lower dimension than the simulated and observed data vectors $\mathbf{z}^*$ and $\mathbf{y}$. In that case, the discrepancy function is zero whenever $s(\mathbf{z}^*) = s(\mathbf{y})$, which can occur even if $\mathbf{z}^* \neq \mathbf{y}$. A practical reason for basing the discrepancy on a vector of summary statistics is that this places weaker constraints on the acceptability of proposed pairs $(\boldsymbol{\theta}^{**}, \mathbf{z}^*)$. If the summary statistics are sufficient for $\boldsymbol{\theta}$, nothing is lost by using $s(\mathbf{y})$ instead of $\mathbf{y}$, but more usually the summary statistics capture much, but not all, of the information $\mathbf{y}$ can reveal about $\boldsymbol{\theta}$.

As a result of these two departures from standard ABC practice, it will not in general be possible for the discrepancy $\rho(\mathbf{z}, \mathbf{y})$ in equation (2.11) to be arbitrarily small. The problem is that since the generative data model does not include a noise term, there may be no parameter vector for which the solution to the system of ODEs exactly corresponds to the observations $\mathbf{y}$, and hence there is some minimum allowable discrepancy $\epsilon_0 > 0$. Consequently, $\rho(\mathbf{z}, \mathbf{y})$ will always be greater than 0, which is considered a misspecification in ABC estimation, according to Frazier *et al.* [36]. Under ideal conditions, for the function $\boldsymbol{\theta} \mapsto \rho(z(\boldsymbol{\theta}), \mathbf{y})$ there is a unique $\boldsymbol{\theta}_0$ such that $\rho(z(\boldsymbol{\theta}_0), \mathbf{y}) = \epsilon_0 > 0$, where $z(\boldsymbol{\theta})$ is the unique solution to the system of ODEs with parameter vector $\boldsymbol{\theta}$. Therefore, as $\epsilon \to \epsilon_0$ from above, the approximate posterior $p_\epsilon(\boldsymbol{\theta}, z \mid \mathbf{y})$ approaches a Dirac delta function at the point $(\boldsymbol{\theta}_0, z(\boldsymbol{\theta}_0))$.

It follows that the approximate 'posterior' $p_\epsilon(\boldsymbol{\theta}, z \mid \mathbf{y})$ targeted by the method of Toni *et al.* [16] contains no information about the posterior variance of parameters. A practical demonstration of this is provided in the results below, in which small to moderate changes in the noise model used to simulate the observations resulted in no change in the posterior variance estimated by ABC methods. On the other hand, changing the noise model used to simulate observations did affect the location of the posterior and the final $\epsilon$ that guaranteed a good acceptance rate.

However, our results presented below demonstrate that the *shapes* of the contours of distributions of the form of equation (2.2) for $\epsilon > \epsilon_0$ may resemble those of the true posterior, and we propose that it may be possible to find some $\epsilon > \epsilon_0$ for which $p_\epsilon$ approximates the true posterior. Finding a good way to do this is left for future work.

# 3. Test problems

The ABC and MCMC techniques described in appendix A were compared against each other when conducting parameter inference for one epidemiological compartmental model. The Bayesian parameter inference software developed in this paper was validated using the method of posterior quantiles [37] on a computationally inexpensive model described in §3.1, before being implemented on a more demanding nonlinear system of ODEs describing malaria transmission in §3.2.

## 3.1. Test problem 1—susceptible–infected–recovered model

Susceptible–infected–recovered (SIR) models categorize hosts into one of three different compartments at time $t$. Individuals are considered susceptible ($S$), if they are able to be infected by the pathogen, infected ($I$) if currently infected with the pathogen or recovered ($R$) if they have successfully cleared the pathogen. The flow of individuals between compartments in the SIR model is visualized in figure 1.

SIR models and their variants, in both deterministic and stochastic forms, are among the most fundamental epidemiological models and have found use describing diseases as diverse as influenza, herpes and malaria [38]. In this test problem we use the SIR model to represent the fraction of the total population ($P$) in each category as follows:

$$s(t) = \frac{S(t)}{P}, \quad i(t) = \frac{I(t)}{P} \quad \text{and} \quad r(t) = \frac{R(t)}{P},$$

where $S(t), I(t)$ and $R(t)$ are the numbers of susceptible, infected and recovered individuals in the population at time $t$ (weeks). The deterministic, constant population, SIR model without demographics can be described mathematically as:

$$\left.\begin{aligned} \frac{\mathrm{d}s}{\mathrm{d}t} &= -\beta i(t)s(t), \\ \frac{\mathrm{d}i}{\mathrm{d}t} &= \beta i(t)s(t) - \gamma i(t) \\ \frac{\mathrm{d}r}{\mathrm{d}t} &= \gamma i(t), \end{aligned}\right\} \tag{3.1}$$

and

where $\beta$ is the infection rate and $\gamma$ is the recovery rate.

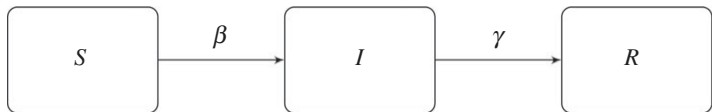

**Figure 1.** The transition of individuals between susceptible, infected and recovered states in an epidemiological compartmental model.

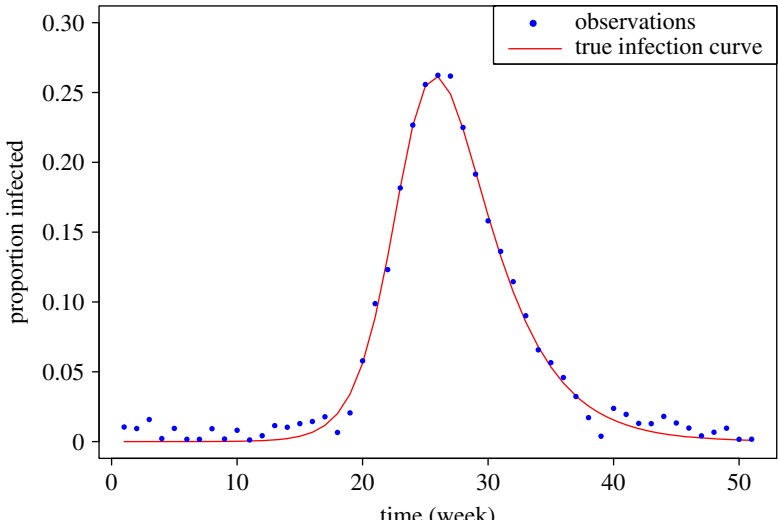

**Figure 2.** Test data showing the proportion of population infected over time, obtained from equation (3.1) with $\beta = 0.9$, $\gamma = 0.333$, $S(0) = 1 - 1.27 \times 10^{-6}$, $I(0) = 1.27 \times 10^{-6}$ and $R(0) = 0$. Red line shows the continuous infection curve while the blue points are the observations to be used to infer the model parameters.

When conducting parameter inference for this system, the observable data is $\mathbf{y} = (y_1, \ldots, y_n)$, where $y_k = i(t_k)$ is the proportion of the population infected at time $t_k$, for $k = 1, \ldots, n$. The parameters of interest are $\boldsymbol{\theta} = \{\beta, \gamma\}$.

### 3.1.1. Simulation results

A test dataset was generated by solving the system of equations (3.1) in the interval $[0, 50]$ using a fourth order Runge–Kutta method and storing the solution at weekly intervals (figure 2), using true model parameters $\boldsymbol{\theta} = (\beta = 0.9, \gamma = \frac{1}{3})^T$. To generate observations $\mathbf{y}$, normal noise $\mathcal{N}(0, \sigma^2 = 0.0001)$ was added to the solution. For ABC approaches, a discrepancy function $\rho(\mathbf{z}, \mathbf{y})$ was used to compare infected proportions in the dataset $\mathbf{y}$ with a solution to the equations $\mathbf{z} = (z_1, \ldots, z_n)$ for proposed parameters as follows:

$$\rho(\mathbf{z}, \mathbf{y}) = \frac{1}{n} \sum_{i=1}^{n} (z_i - y_i)^2, \tag{3.2}$$

where $n$ is the number of observed data points. The priors for $\beta$, $\gamma$ and $\sigma^2$ were taken to be vague:

$$\beta \sim \mathcal{U}(0, 2), \quad \gamma \sim \mathcal{U}(0, 2) \quad \text{and} \quad \sigma^2 \sim \mathcal{IG}(1, 1) \tag{3.3}$$

where $\mathcal{U}(\cdot, \cdot)$ is the uniform distribution and $\mathcal{IG}(\cdot, \cdot)$ is the inverse-gamma distribution. For MCMC approach, normal proposal distributions were used with adaptive approach tuning parameters in the algorithm to maintain an acceptance ratio between 0.3 and 0.5 [25].

Given the observations $\mathbf{y}$, the parameter vector $\boldsymbol{\theta} = \{\beta, \gamma\}$ was estimated using MCMC and SMC ABC and the results from these methods were compared. The noise $\sigma^2$ was additionally estimated when using MCMC. As discussed in §2.2, the distributions derived using the ABC approaches are not an approximation to the true posterior of the ODE model parameters since the noise is not estimated; we therefore cannot use the standard deviation of the distributions from ABC approaches as a measure of performance. Instead, we compared the CPU times, the number of iterations and the mean absolute

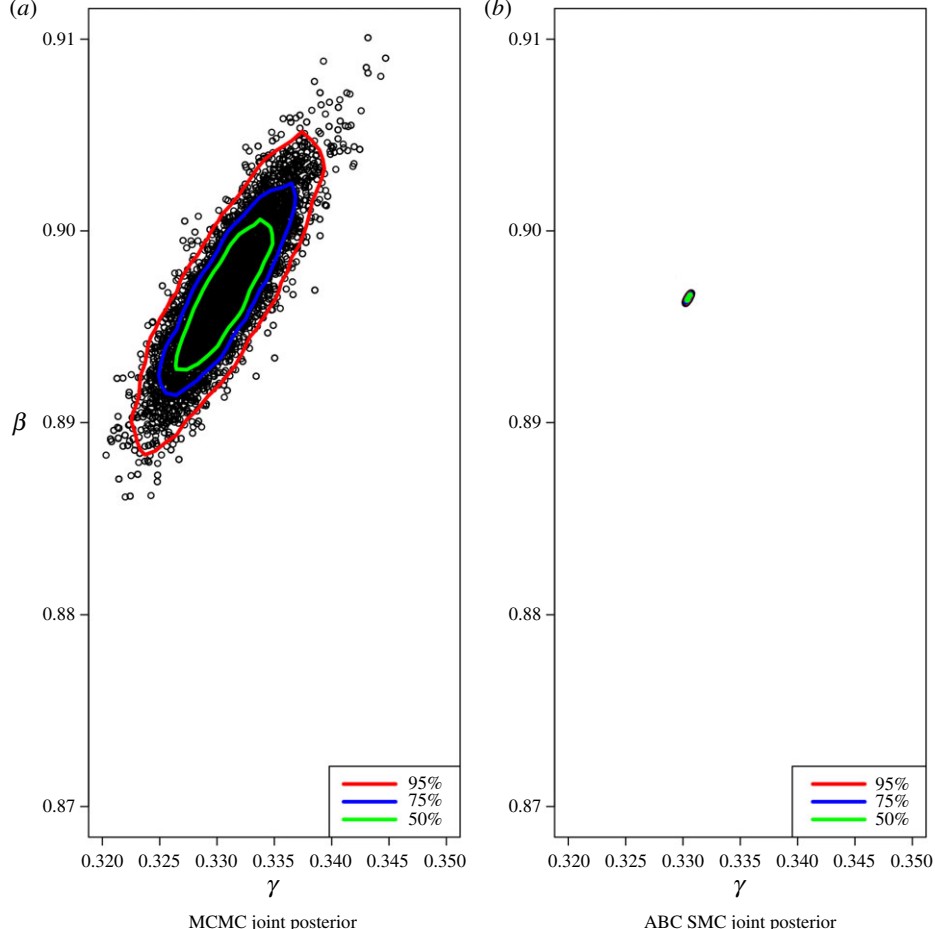

**Figure 3.** Scatter plot of sample draws for $\gamma$ and $\beta$ using MCMC (*a*) and SMC ABC (*b*). The contour lines contain the stated proportions of sample draws from the joint posterior and are produced using the R function 'HPDregionplot'.

**Table 1.** The number of iterations, computational time (min) and mean absolute error for parameter inference in the SIR model.

|          | iterations | CPU time | MAE ($\beta$) | MAE ($\gamma$) |
| -------- | ---------- | -------- | ------------- | -------------- |
| MCMC     | 12 401     | 6.58 min | 0.0038        | 0.0034         |
| SMC ABC  | 141 408    | 11.29 min | 0.0035       | 0.0029         |

errors (MAE), although, MAE may favour over-concentrated posterior approximations. The formula used for MAE is $\text{MAE} = \sum_{i=1}^{n} (|\theta_i - \theta_{\text{true}}|)/n$, and the $\theta_i$ are posterior samples for $\theta$, for each method. We applied the MCMC approach using appendix A, algorithm 2. Table 1 shows that MCMC chain converged after 12 401 steps to reach convergence and this took approximately 6.58 min.

We next applied SMC ABC as outlined in appendix A, algorithm 6 using our own implementation in R. In SMC ABC code we used $T = 11$ populations, each with 1000 particles, used component-wise uniform kernels that adapted their width from the previous particle distributions [28] and used uniform priors 2 units wide and centred at zero for both parameters. The tolerance sequence was selected adaptively such that in population $i$ the new threshold $\epsilon_i$ was the 25th percentile of the distances in the previous iteration, $t-1$ (as explained in §2). The algorithms terminated when we reached a challenge tolerance of $\epsilon = 0.067056$ that had been chosen by finding the distance between the true ODE solution and the generated observations **y**.

Comparing SMC ABC with MCMC, we found that SMC ABC consumed run times longer than MCMC with 11.26 min. In addition, It can be seen in figure 3*b* that the estimated joint posterior resulting from the 11th population of the SMC ABC method has the smaller variance compared with MCMC method. Table 2 shows that all the methods have achieved good point estimation for both parameters $\beta$ and $\gamma$.

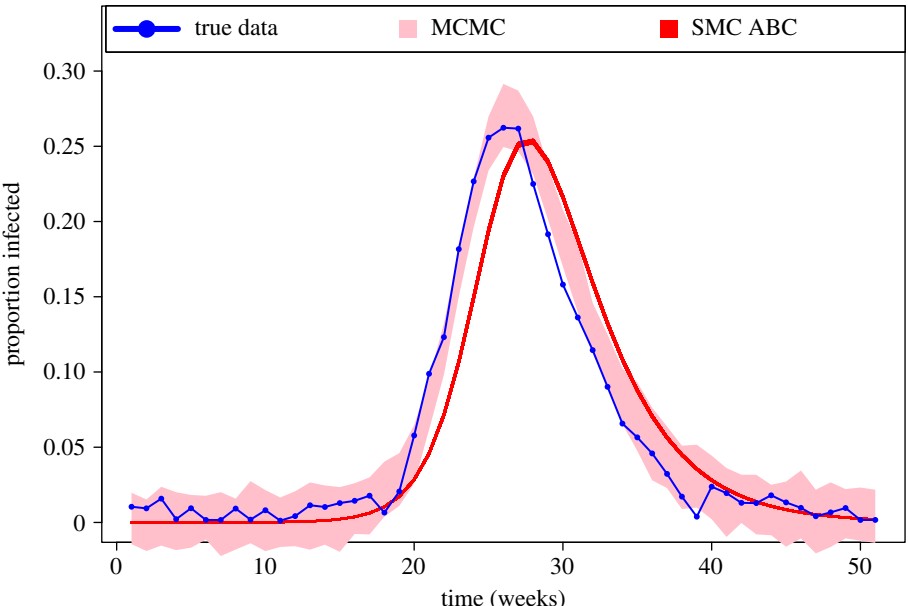

**Figure 4.** Plot of the posterior predictive credible intervals estimated using MCMC and SMC ABC fitted with the weekly infection cases. The blue dots represent the noisy data. The shaded areas are created using the posterior predictive samples. It can be seen that the result derived from MCMC covered most of the data points while the ABC derived result produces unrealistically narrow credible intervals.

**Table 2.** True values of the parameters $\beta$ and $\gamma$ and their estimated values (each estimate is the median of the sampled values) using MCMC and SMC ABC for SIR model example.

| parameter | true value | MCMC | SMC ABC |
|---|---|---|---|
| $\beta$ | 0.9 | 0.8968 | 0.8964 |
| $\gamma$ | 0.3333 | 0.3308 | 0.3304 |

We also compared credible intervals for the solution to the system of ODEs, using MCMC and ABC. To do this, we followed the procedure in Gelman & Rubin [25] to simulate the posterior predictive distribution (PPD) for a future observation, $\mathbf{y_{rep}}$. This produces a PPD that has variance that depends on the posterior uncertainty (and hence the observational noise). This was not a problem within MCMC because we take samples from the posterior and solve the model and then simulate the noise, which has been estimated within MCMC. However, with SMC ABC method, we take the samples from the posterior and solve the model without simulating the noise. So, as the tolerance gets smaller and smaller the predictive intervals will also get smaller, and do not have the correct coverage of the observed data. Figure 4 shows that the credible intervals obtained using SMC ABC are much narrower than those obtained using MCMC, highlighting that the variation within the sample does not contain useful information about the inferred uncertainty of the estimates. Note that this problem is not reduced by the tunable elements of the algorithm; for example, we tried several different perturbation kernels proposed by Filippi *et al.* [28], such as component-wise perturbation kernels that adaptively choose width based on the previous population and multivariate normal perturbation kernels that are sometimes useful when parameters are highly correlated. The resulting credible intervals were not affected significantly by the choice of perturbation kernel (comparison results not shown).

In addition, we used the proposed method in Prangle [33] within our R code, involving an adaptive distance function to improve the performance of the SMC ABC method. In this algorithm, the scale parameters $\zeta_j$ in the distance function (equation (2.4)) are updated in each iteration (calculated using MAD) and are used to choose the value of the next $\epsilon_t$. In principle, it might be possible to use a variant of this technique to choose $\epsilon_t$ so that the resulting sample reflects the shape and spread of the true posterior distribution. However, it is not clear what number of rounds of adaptation would produce such an $\epsilon_t$. In other words, it is not clear how and when to terminate the SMC ABC algorithm. In the literature, there

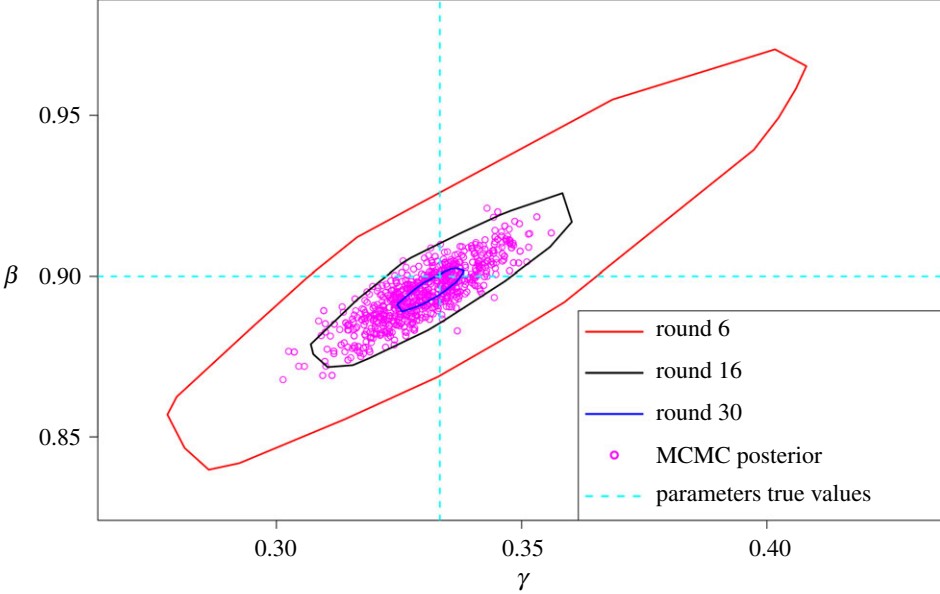

**Figure 5.** An illustration of SMC ABC for the SIR model using algorithm 4 in Prangle [33]. The red curve represents the posterior estimate resulting from the sixth round and is clearly overdispersed (by comparison to the sample obtained using MCMC—purple scatter points). The black curve, obtained after 16 rounds of SMC ABC, is the closest approximation to the posterior obtained using SMC ABC. The blue curve is the posterior estimate obtained after 30 rounds of SMC ABC, and has clearly shrunk too much around the true parameter values (dashed light blue lines).

are several methods one can use to terminate, such as when the algorithm reaches a certain value of $\epsilon$ or a target acceptance rate, or one can use a specified number of a total simulations as a tuning parameter and the algorithm terminates when a further simulation is required [33]. One option that is available when using an adaptive kernel width is to stop the algorithm when the width of the kernel becomes negligible or when $1 - (\epsilon_{t+1}/\epsilon_t)$ falls below some threshold. However, we found that none of these methods terminate the algorithm in such a way as to produce the correct shape and spread of the posterior distribution. Figure 5 illustrates that if we run the algorithm for 30 rounds the estimated posterior will shrink towards a point estimate of the parameters. If we run the algorithm for six rounds, the resulting estimate is not a good representation of the true posterior: it is too wide and hence misleading. An estimated posterior distribution similar to what we obtain using MCMC can be somewhat artificially generated if we run the SMC ABC algorithm for 16 rounds, but at present the algorithm lacks an independent way of identifying this. As far as we are aware, there are as yet no clear guidelines to follow to determine how many rounds of SMC ABC are needed when dealing with an ODE model to guarantee that a good approximation to the true posterior of the parameters has been achieved.

### 3.1.2. The impact of the noise on the inference

To demonstrate the impact of the noise on the parameter inference we applied MCMC and SMC ABC to observations **y** generated using different values of $\sigma^2$, specifically $\sigma^2 \in \{0.0001, 0.0005, 0.001\}$. We plot the resulting posterior marginals for both parameters in figure 6. The variance of the estimated posterior derived from MCMC increases as the value of $\sigma^2$ increases, indicating that the variance of the posterior is affected by the amount of noise, as expected. On the other hand, although the noise has been increased, the variances of the estimated posteriors derived from SMC ABC are almost the same for small to moderate amounts of noise. However, when the noise parameter is increased further, the location of the estimated posterior is changed. This illustrates what has been discussed in §2.2 that these posteriors do not provide valid information about the uncertainty in the parameter estimates. As a result, conducting parameter estimation for ODE models using this ABC framework is not recommended.

### 3.1.3. Including the error term in the ABC algorithm

Including the error term in the ABC algorithm may overcome this limitation and work inspired by Wilkinson [35] in Vaart *et al.* [34], as has been explained in §2, argued that the acceptance of the

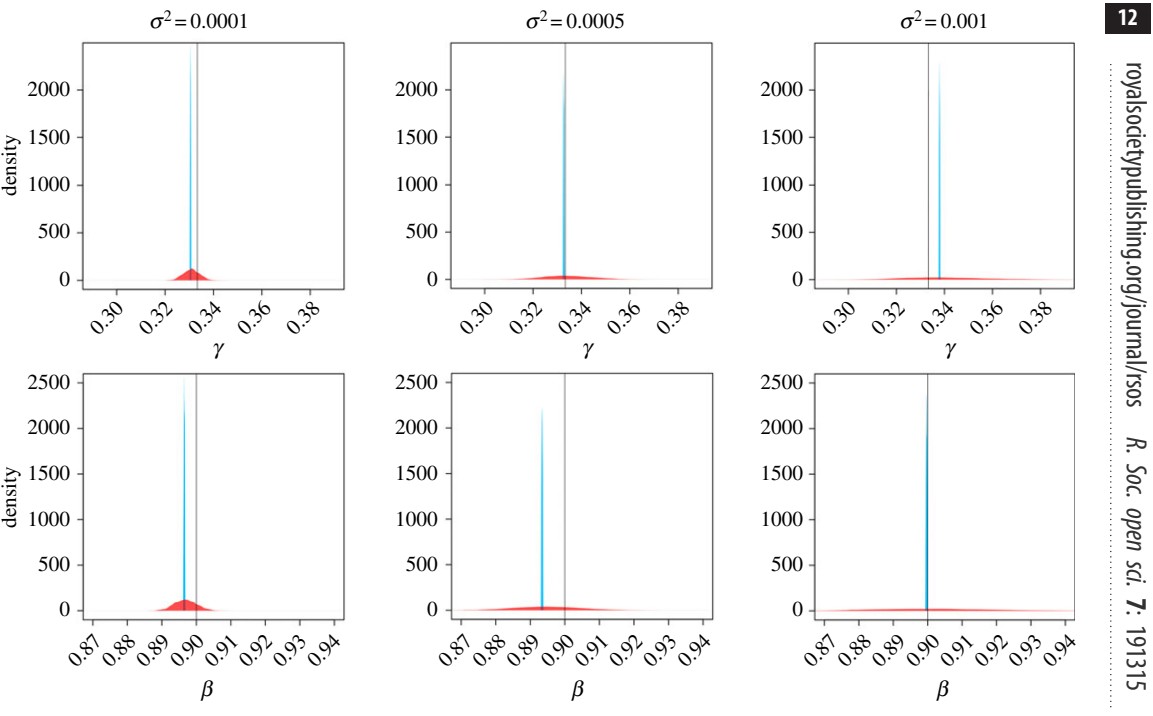

**Figure 6.** Plots of the estimated posterior marginal densities for parameter $\beta$ and $\gamma$ obtained using MCMC (red) and SMC ABC (blue) with different amounts of noise. The black solid line represents the true value of the parameters. It is clear that the variance of the posteriors derived from MCMC is affected by increasing the noise, but this is not the case for posteriors obtained using SMC ABC.

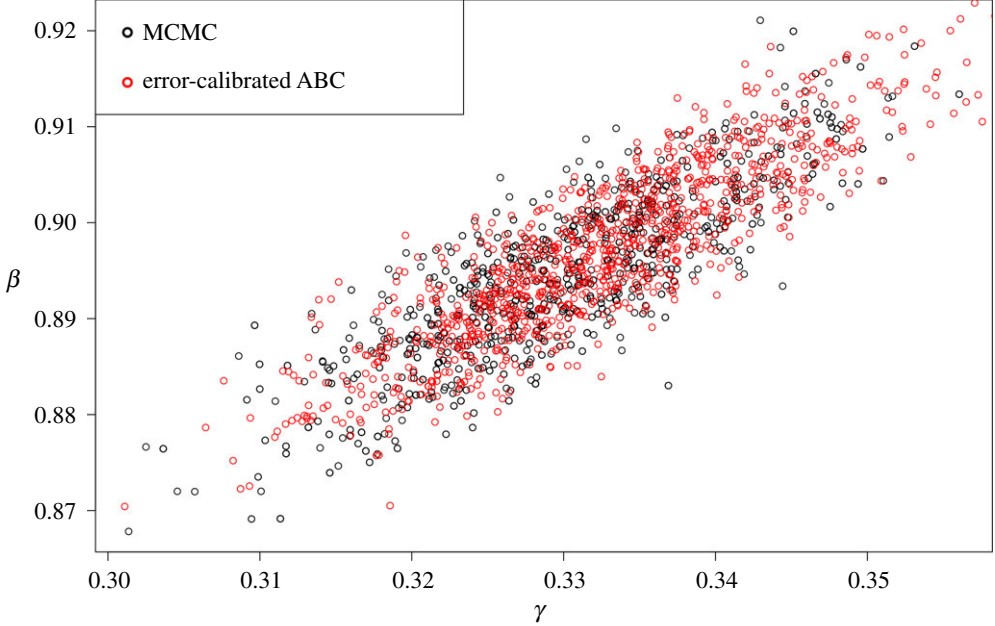

**Figure 7.** Scatter plot of posterior distribution sample draws for $\beta$ and $\gamma$ from MCMC (black) and error-calibrated ABC (red) for the SIR model in SIR model example.

proposed parameters should be with respect to the error term rather than with respect to some tolerance level. In their method they assumed that the error term follows a normal distribution. This method is promising and can capture similar posterior shapes compared to the one derived from MCMC, as figure 7 shows. A significant drawback that we found for this approach is that the acceptance rate is very

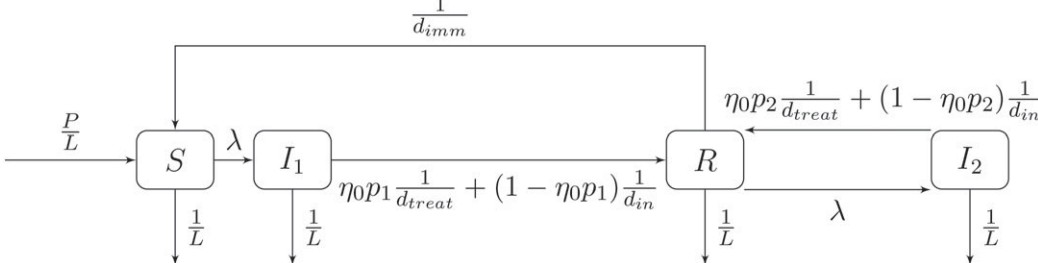

**Figure 8.** The flow of individuals between susceptible, infected and recovered states in the model of White *et al.* [39].

low and a large number of simulations are needed. This leads to a longer computational time, which is prohibitive for the case of ODE model parameter inference. The total CPU time when we applied the Vaart *et al.* [34] algorithm to the first test problem to have 1000 samples is 29 h which was derived from $2 \times 10^6$ simulations and this time is much larger than when we used the MCMC method for this example (5.25 min).

## 3.2. Example 2—nonlinear ODE model of malaria transmission

Work by White *et al.* [39], acknowledging the lack of reliable data in some countries where malaria control or elimination is particularly desirable, showed the utility of a compartmental mathematical model in predicting effects of various elimination strategies compared to the more complex models of Gu *et al.* [40] and Maire *et al.* [41]. The model describes population dynamics using four population compartments in the transmission of malaria:

$S(t)$: Uninfected and non-immune.
$I_1(t)$: Infected with no prior immunity.
$R(t)$: Uninfected with immunity.
$I_2(t)$: Infected with prior immunity.

The model comprises four ODEs that govern the temporal evolution of the population compartments. The model is illustrated in figure 8 and can be described mathematically by the following equations:

and
$$\left.\begin{aligned}
\frac{dS}{dt} &= \frac{P}{L} - \left(\lambda + \frac{1}{L}\right)S + \frac{1}{d_{\text{imm}}}R, \\
\frac{dI_1}{dt} &= \lambda S - \left(\frac{\eta_0 p_1}{d_{\text{treat}}} + \frac{1 - \eta_0 p_1}{d_{\text{in}}} + \frac{1}{L}\right)I_1, \\
\frac{dI_2}{dt} &= \lambda R - \left(\frac{\eta_0 p_2}{d_{\text{treat}}} + \frac{1 - \eta_0 p_2}{d_{\text{in}}} + \frac{1}{L}\right)I_2 \\
\frac{dR}{dt} &= \left(\frac{\eta_0 p_1}{d_{\text{treat}}} + \frac{1 - \eta_0 p_1}{d_{\text{in}}}\right)I_1 + \left(\frac{\eta_0 p_2}{d_{\text{treat}}} + \frac{1 - \eta_0 p_2}{d_{\text{in}}}\right)I_2 - \left(\lambda + \frac{1}{d_{\text{imm}}} + \frac{1}{L}\right)R.
\end{aligned}\right\} \tag{3.4}$$

Here $\lambda$ is the force of infection and is given by

$$\lambda = R_0\left(\frac{1}{L} + \frac{1}{d_{\text{treat}}}\right)\frac{(I_1 + I_2)}{P},$$

where $R_0$, the average number of secondary infections arising from a single infected individual in a susceptible population, is expressed as a function of time to incorporate the seasonal forcing associated with malaria transmission and is of the form, $R_0(t) = A\cos 2\pi(t - \phi) + r_0$. The model is parametrized in terms of a number of constants as described in table 3.

The observed data, $\mathbf{y(t)}$, is taken to be the number of observable clinical infections, $C(t)$ as follows:

$$\mathbf{y(t)} = \{C_1(t_1), C_2(t_2), \ldots, C_{n-1}(t_{n-1}), C_n(t_n)\} \tag{3.5}$$

and

$$C_n(t_n) = p_1 \mathbf{I_1}(t_n) + p_2 \mathbf{I_2}(t_n). \tag{3.6}$$

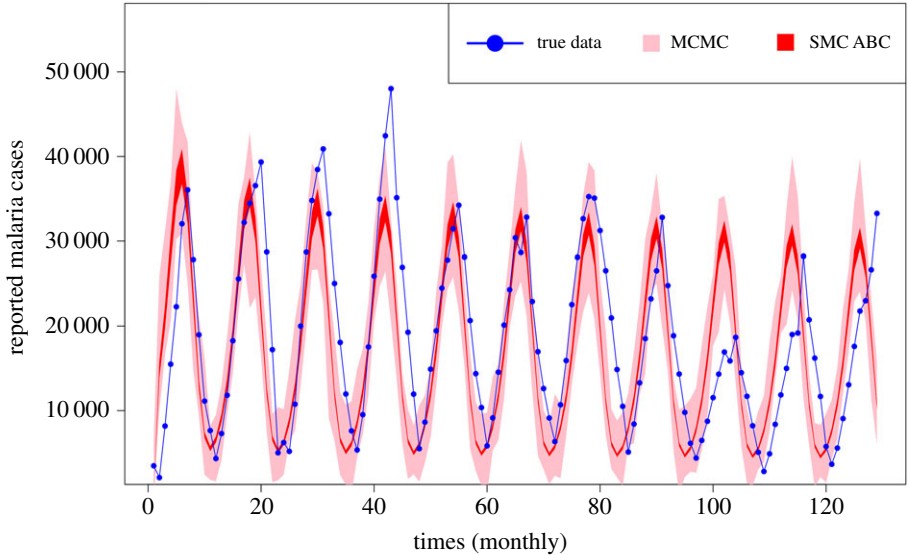

**Figure 9.** Plot of the posterior predictive credible intervals from MCMC and SMC ABC fitted with the monthly Afghanistan data. The blue dots represent the data. The shaded areas are created by the posterior predictive samples. The result derived from MCMC covered most of the data points while the ABC methods were unable to cover most of the data.

**Table 3.** The parameter values used in simulation of the White *et al.* [39] model.

| parameter | | value | source |
|---|---|---|---|
| $P$ | (people) | 29 203 486 | Worldometers [42] |
| $L$ | (years) | 66.67 | Maude *et al.* [43] |
| $d_{imm}$ | (years) | 0.93 | Aguas *et al.* [44] |
| $d_{in}$ | (years) | 0.11 | assumed |
| $d_{treat0}$ | (weeks) | 2 | Maude *et al.* [43] |
| $p_1$ | | 0.87 | Aguas *et al.* [44] |
| $p_2$ | | 0.08 | Aguas *et al.* [44] |
| $A$ | | 0.67 | assumed |
| $r_0$ | | 1.23 | assumed |
| $\phi$ | | 3/12 | assumed |
| $\eta_0$ | | 0.11 | assumed |

For our purposes here, the parameter vector of interest is $\boldsymbol{\theta} = (\eta_0, d_{in})$ and in the case of using MCMC, the parameter vector of interest is $\boldsymbol{\theta} = (\eta_0, d_{in}, \sigma^2)$, where $\eta_0$ is the percentage of individuals with clinical infection that receive treatment, $d_{in}$ is the average duration of an untreated sensitive infection and $\sigma^2$ is the noise associated with the data which we assumed to be normally distributed.

### 3.2.1. Application: malaria in Afghanistan

Afghanistan is a landlocked country located between South Asia and Central Asia. Despite the fact that most of the country is desert, there is significant rainfall and snowfall [45], which provides a fertile environment for mosquito-borne diseases such as malaria. We use monthly data from cases registered nationwide across all regions of Afghanistan in the period from January 2005 to September 2015 from Anwar *et al.* [46] as shown in figure 9.

In the ODE system in equation (3.4), $I_1$ and $I_2$ represent the number of infected individuals with no prior immunity and prior immunity, respectively. However, in the case of the data from Afghanistan, each data point represents the total number of malaria cases that arrived at hospitals in the month. In order to calculate

**Table 4.** The estimated values (the median of the posterior) of the parameters $\eta_0$ and $d_{in}$ from MCMC, MCMC ABC and SMC ABC for the Afghanistan data.

| parameter | MCMC | MCMC ABC | SMC ABC |
|---|---|---|---|
| $\eta_0$ | 0.0525 | 0.04685 | 0.0459 |
| $d_{in}$ | 0.23035 | 0.2483 | 0.2453 |

**Table 5.** The number of iterations and computational time (min) for parameter inference in the malaria model, applied to the Afghanistan data.

| | iterations | CPU time |
|---|---|---|
| MCMC | 5838 | 45.13 min |
| MCMC ABC | 4050 | 39.99 min |
| SMC ABC | 143 031 | 522.34 min |

the cumulative number of cases over time we added an extra ODE which takes the form:

$$\frac{dW}{dt} = \lambda S \eta_0 p_1 + \lambda R \eta_0 p_2, \tag{3.7}$$

where $W(t)$ is the cumulative number of observed (that is, treated) cases. To compute the number of new cases in each month we subtract the cumulative cases from consecutive months.

The values of the model parameters used are shown in table 3 and the initial conditions are given by the equilibrium solution of the system in equation (3.4) with the addition of $W(t=0)=0$ for the new ODE. As with the first test problem, for the ABC approaches we used the discrepancy function in equation (3.2) to compare the clinical infections given in the dataset **y** with a simulated solution **x**. The priors for $\eta_0$, $d_{in}$ and $\sigma^2$ were taken as follows:

$$\left.\begin{array}{l} p(\eta_0) = \mathcal{B}(1, 1) \in [0, 1], \\ p(d_{in}) = \mathcal{GA}(1, 1) \in [0, \infty) \\ p(\sigma^2) = \mathcal{IG}(1, 1) \in [0, \infty). \end{array}\right\} \tag{3.8}$$

and

A logistic transformation was used to transform $\eta_0 \in [0, 1]$ while a log transform was applied to $d_{in} \in [0, \infty)$ and $\sigma^2 \in [0, \infty)$ so that each transformed parameter had support over the real line. This step was used to improve the acceptance rate of the proposals.

In MCMC method, zero mean normal proposal distributions were used with standard deviations equal to (0.007, 0.07, 0.1) for the parameters ($\eta$, $d_{in}$, $\sigma^2$) respectively. The same proposal distributions were used with MCMC ABC, but with standard deviations equal to (0.1, 0.1) for the parameters ($\eta$, $d_{in}$).

All of the methods have achieved convergence to similar values for both parameters under investigation ($\eta_0$, $d_{in}$), as shown in table 4. SMC ABC consumed significantly longer CPU time compared with the other methods as shown in table 5. Also, SMC ABC needed 143 031 model simulations to get 500 accepted values, while MCMC and MCMC ABC needed just 5838 and 4050 iterations, respectively to converge (tables 4 and 5).

Since a real dataset has been used here, the true parameter values are unknown. As a consequence, applying MCMC ABC was difficult because this lack of information makes the choice of an appropriate $\epsilon$ problematic. In this paper, in order to select an appropriate tolerance level we adopted the method of Vaart et al. [34], which is to solve the ODE model with different proposals of the parameters from the priors, find all the distances using 3.2 between these solutions and the true data, and then choose the one that minimizes this distance. We then used the best fitting solution to estimate the value of the MCMC ABC tolerance which was $\epsilon = 116230.8$. Then, we applied SMC ABC for six populations with an adaptively chosen sequence of tolerance $\epsilon = (244616.4, 244616.4, 244616.4, 176677.1, 116966.8, 100042.7)$.

The estimated joint posteriors of $\eta_0$ and $d_{in}$ can be seen in figure 10. All have the same shape and similar position, but the variances are very different. Figure 9 shows that the posterior predictive distribution from MCMC covers most of the data points; however, the predictive intervals from ABC methods (here, showing only the SMC ABC result from the last population) are very tight and poorly

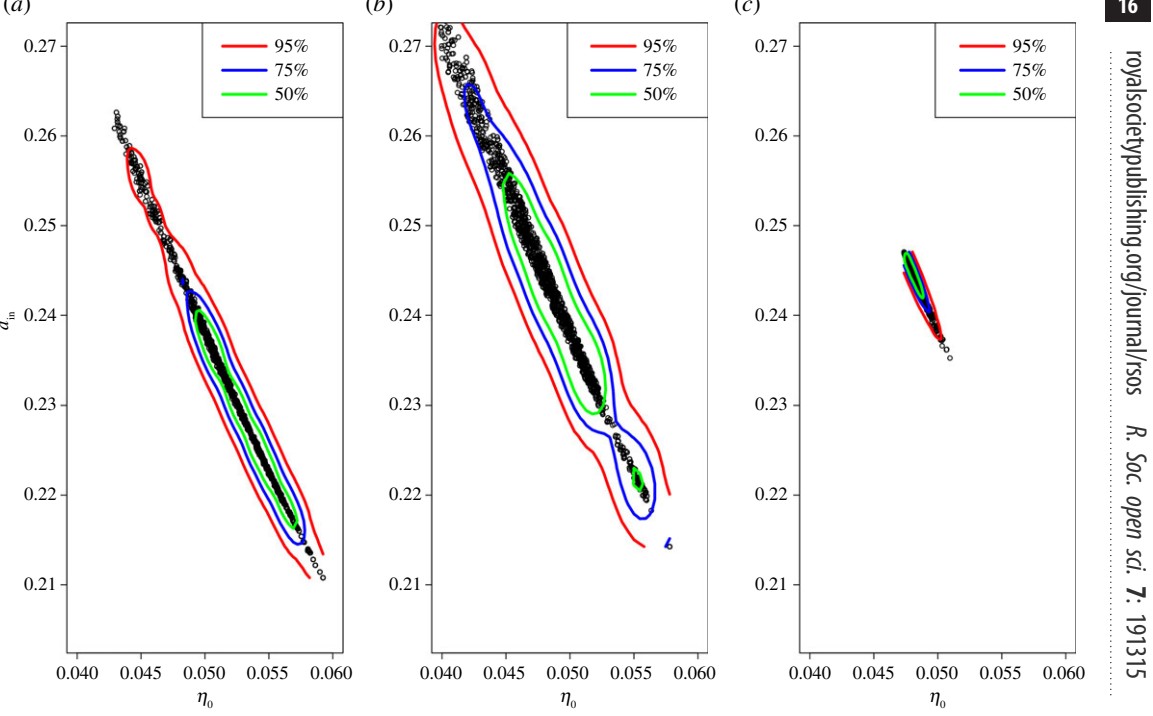

**Figure 10.** Scatter plot of posterior distribution sample draws for $\eta_0$ and $d_{in}$ obtained using MCMC (a), ABC MCMC (b) and SMC ABC (c) for model applied to Afghanistan data. The contour lines contain the stated proportion of sample draws and they were produced using the R function 'HPDregionplot'.

cover the data points. The reason for this is, as the SMC ABC algorithm converges towards the point estimate, without any consideration to the noise in the data, the tolerance gets smaller and then most of the accepted parameter values are from a tight region around a point estimate (figure 10). Thus, the predictive intervals from ABC methods do not enable appropriate coverage of the data, consistent with the discussion in §2.2.

## 4. Discussion

Our investigation into exact and approximate methods for inferring parameters in ODE dynamical systems under a Bayesian framework highlights some limitations of current methods. The main problem we identified is that the observations of the system are often noisy, so when we infer the parameters for such a system it is inappropriate to not simulate the noise process. Estimation of the noise parameter is standard using exact Bayesian inference (MCMC), but not with the current practice with ABC-based approaches when applying to a system of ODEs that we investigated here. The general idea when applying the ABC-based methods considered here to a dynamical system is to compare the noisy observations to solutions generated using the ODEs (which is a deterministic model that does not take into consideration the noise in the observations) for each set of parameters proposed. The parameters are accepted based on some tolerance $\epsilon$ that also does not depend on the noise term.

To illustrate this limitation, we compared the popular methods MCMC, MCMC ABC and SMC ABC for estimating model parameters in ODEs. We can see in the second example presented in this paper that the computational time consumed by MCMC ABC is shorter compared to the other methods (MCMC and SMC ABC). However, when dealing with a deterministic model, the estimated posteriors derived from current ABC methods do not provide useful estimates of the true posteriors. In particular, they do not contain appropriate information about the uncertainty of the parameter values. Being able to naturally quantify uncertainty in posterior distributions is one of the main advantages of Bayesian statistical inference over other approaches, given that the output of Bayesian inference is a probability distribution rather than a point estimate. Here, we have shown that the ABC methods are not able to capture the distribution, but instead converge to point estimates of the best parameter values. To

demonstrate the effect of not including the noise term in the estimation we compared the results in the first example for different noise values. It is clear that the distribution estimated using the ABC methods is not affected by the noise values (for low to moderate amounts of noise), which illustrates that the posterior variances estimated using ABC do not depend on the noise in the data. As a result, the posteriors do not represent the whole probability distribution of the parameters under investigation.

In order to improve SMC ABC we tried an adaptive SMC ABC on the first test problem by using several different perturbation kernels that are proposed in Filippi *et al.* [28]; component-wise perturbation kernels and multivariate normal perturbation kernels that adapt their width from the previous population in the algorithm steps. Adapting the width of the kernels may affect the accuracy of the estimations, but still does not capture the true posterior shape when comparing with the posterior obtained using an MCMC method.

We also tried another adaptive SMC ABC method in this paper; an adaptive distance function as in §2, proposed in Prangle [33]. This method did make it possible to obtain a distribution that more closely approximates the posterior using SMC ABC, but the problem with this algorithm is that there is no existing criteria to identify an appropriate iteration at which to terminate.

Including the error term in the ABC algorithm may improve the ABC posterior as we have seen when we applied Vaart *et al.* [34] method in §2; however, the long computational times required by this approach are considered as a remarkable drawback in the case of ODE model parameter inference.

The comparison conducted in this paper demonstrates that using exact Bayesian inference (MCMC) for ODE parameter estimation is a practical alternative (Gelman *et al.* [47]), despite the difficulty involved in calculation of the likelihood. We found that MCMC gave accurate estimation of the parameter values and the resulting posterior gave appropriate information about the uncertainty of the parameters. Furthermore, the variance of the MCMC posterior distribution changed as the noise in the data changed, as one would expect. The same was not true for the ABC methods considered. The time consumed by the MCMC algorithms was slightly larger than MCMC ABC; however, since the resulting posteriors were more appropriate, the extra effort to calculate the likelihood is deemed worthwhile. In addition, choosing an appropriate $\epsilon$ when applying MCMC ABC is difficult, especially when working with real data. With simulated data, it is possible to find an appropriate $\epsilon$ from the distance between the true solution of the ODE model and the noisy data, but this is not possible in a real application where the true solution is unknown. In this case, in order to determine an acceptable tolerance level, we adapt the work of Vaart *et al.* [34] to find the best fit solution and then find an appropriate $\epsilon$. We found that among all the methods, applying SMC ABC is the easiest to implement, but consumes the most computational time. Moreover, as we have observed, this method produces inappropriately shaped posterior distributions.

The first example presented in this paper involves a likelihood function that is easy to compute, so using a likelihood-based approach such as MCMC or an importance sampling method like SMC is certainly to be preferred over likelihood-free methods (such as ABC). Most of the computational cost of MCMC and SMC ABC method is consumed in solving the ODE models several times to compute the likelihood for MCMC or to do the simulations for SMC ABC. However, in the second example we found that more effort is needed to construct the likelihood functions when applying MCMC. In addition, when we chose an uninformative prior for the parameters, the SMC ABC algorithm located the appropriate region of the parameters space easily, while it was more difficult to choose appropriate initial parameters to achieve rapid convergence with MCMC. We would currently recommend users of ABC methods be careful when using it with ODEs, unless a sensible choice of error model and summary statistics can be made. Deciding what are sensible choices for the ABC algorithm is still difficult and an important topic of current and future work.

Data accessibility. All the code and the data required to reproduce the results presented in this paper is available as electronic supplementary material and appendix.

Authors' contributions. A.A.A. produced the final codes of the study, performed the analysis, interpreted results and wrote the paper. D.G.C. participated in developing the initial draft of the code and the paper. C.C.D. provided guidance on the ABC implementation and revised the manuscript. J.A.F. and J.M.K. conceived of the presented idea, supervised the findings of this work and revised the manuscript.

Competing interests. We declare we have no competing interests.

Funding. The authors are grateful to the Australian Research Council Centre of Excellence for Mathematical and Statistical Frontiers for their support of this project no. CE140100049.

Acknowledgements. We thank Prof. Richard Wilkinson, the editors and the six anonymous reviewers, whose comments helped us improve our work.

# Appendix A

## A.1. Markov chain Monte Carlo

MCMC techniques as developed by Metropolis *et al.* [1] and Hastings [2] can be used to sample from the posterior distribution in equation (2.1). The Metropolis–Hastings algorithm constructs a Markov chain for which the stationary and limiting distribution is the posterior distribution. After running the chain for a sufficient amount of time, samples from the chain can be considered draws from the posterior distribution. An implementation of the Metropolis–Hastings algorithm is given in algorithm 2. Algorithm 2 gives no restrictions on the proposal distribution $q(\theta^* | \theta_{t-1})$. An appropriate proposal distribution should allow the entire parameter space to be explored while maintaining an appropriate acceptance rate[2] [48]. For example, $q(\theta^* | \theta_{t-1})$ can be chosen to be $\mathcal{N}(\theta^* | \theta_{t-1}, \Sigma)$ (the multivariate normal distribution with mean vector $\theta_{t-1}$ and covariance matrix $\Sigma$) or to be independent of the previous parameter draw, $\theta_{t-1}$.

---

**Algorithm 2.** The Metropolis–Hastings algorithm [1,2].

---

1: Initialize $\theta_0$.

2: **for** $t = 1$ *to* $T$ **do**

3:      Propose $\theta^*$ from a proposal distribution $q(\theta^* | \theta_{t-1})$.

4:      Calculate $\alpha(\theta_{t-1} \rightarrow \theta^*) = \min\left(1, \dfrac{p(\mathbf{y}|\theta^*)p(\theta^*)q(\theta_{t-1}|\theta^*)}{p(\mathbf{y}|\theta_{t-1})p(\theta_{t-1})q(\theta^*|\theta_{t-1})}\right).$

5:      Set $\theta_t = \theta^*$ with probability $\alpha$, else set $\theta_t = \theta_{t-1}$.

6: **end for**

---

## A.2. Approximate Bayesian computation

MCMC methods require the computation of the likelihood function, $p(\mathbf{y} | \theta)$, in equation (2.1). ABC methods were developed to sample from an approximation to the posterior in cases for which the likelihood is intractable or too computationally costly to compute. Instead of calculating the likelihood, a comparison is made between the observed data, $\mathbf{y}$, and simulated data, $\mathbf{z}$. In general, The simplest ABC algorithm involves the following steps:

1.  Sample a parameter, $\theta^*$, from the prior, $p(\theta)$.
2.  Simulate a dataset, $\mathbf{z}$, from the model $f(\mathbf{z} | \theta^*)$.
3.  Compare the simulated dataset, $\mathbf{z}$, with the observed data, $\mathbf{y}$, using a discrepancy function, $\rho(\mathbf{z}, \mathbf{y})$. If $\rho(\mathbf{z}, \mathbf{y}) \leq \epsilon$, where $\epsilon$ is the desired tolerance level, accept $\theta^*$.
4.  Repeat steps 1–3 until a desired number of samples are accepted.

ABC targets an approximate posterior [26]:

$$p_\epsilon(\theta, \mathbf{z}|\mathbf{y}) \propto \mathbb{1}(\rho(\mathbf{z}, \mathbf{y}) \leq \epsilon)p(\theta)f(\mathbf{z}|\theta), \tag{A 1}$$

where $\mathbb{1}$ is an indicator function that takes the value one if its logical argument is true and zero otherwise. The accuracy of ABC approaches depends on choosing a suitable discrepancy function $\rho(\mathbf{z}, \mathbf{y})$ and an appropriate tolerance $\epsilon$ [9]. In practice, the discrepancy function typically compares sets of summary statistics $s(\cdot)$ for the observed and simulated datasets.

### A.2.1. ABC rejection sampling

The simplest ABC algorithm is ABC rejection sampling [7]. It can be implemented in one of two ways; pre-specification of $\epsilon$, as shown in algorithm 3, or post-determination of $\epsilon$, as shown in algorithm 4. Pre-specification of $\epsilon$ is problematic, since if $\epsilon$ is too small then many simulations will be required to produce

---

[2]The asymptotically optimal acceptance for a random walk Metropolis–Hastings algorithm is 0.44 (in one dimension) or 0.23 if more than one parameter is to be estimated.

$N$ samples and if $\epsilon$ is too large, accuracy will be reduced. How the parameter $\epsilon$ is set represents a trade-off between speed and accuracy.

---

**Algorithm 3.** ABC rejection algorithm, pre-specification of $\epsilon$

1: **while** number of accepted $\theta^* < N$ **do**
2:     Draw $\theta^* \sim p(\theta)$.
3:     Simulate $\mathbf{z}^*$ from model given $\theta^*$.
4:     **if** $\rho(\mathbf{z}^*, \mathbf{y}) \leq \epsilon$ **then**
5:         Accept $\theta^*$.
6:     **end if**
7: **end while**

---

Alternatively $\epsilon$ could be specified after drawing $T$ samples from the prior as demonstrated in algorithm 4. We define a constant, $\alpha$, which represents the percentage of draws that are to be accepted and set $\epsilon$ according to our pre-defined $\alpha$. Although this scheme provides flexibility in terms of trading accuracy for speed, it can require large storage requirements if $T$ is large.

---

**Algorithm 4.** ABC rejection algorithm, post-determination of $\epsilon$.

1: **for** $i = 1$ *to* $T$ **do**
2:     Draw $\theta_t \sim p(\theta)$ and simulate $\mathbf{z}_t$ from model given $\theta_t$.
3:     Compute discrepancy function $\rho_t = \rho(\mathbf{z}_t, \mathbf{y})$.
4: **end for**
5: Sort $\{\theta_t, \rho_t\}_{t=1}^{T}$ into ascending order, based on $\rho$.
6: Keep $N = \alpha T$ of $\theta_t$ with the lowest discrepancy, hence defining $\epsilon$.

---

### A.2.2. MCMC ABC

ABC rejection sampling is very simple to implement, though it can suffer from extremely low acceptance rates when the prior distribution is dissimilar to the posterior distribution [9]. To counteract this deficiency, a more efficient ABC technique based on MCMC was developed [9]. The implementation

---

**Algorithm 5.** MCMC ABC with early rejection [49].

1: Obtain $\theta_0$ and $\mathbf{z_0}$ using ABC rejection sampling.
2: **for** $t = 1$ *to* $T$ **do**
3:     Draw $\theta^* \sim q(\theta^* | \theta^{t-1})$.
4:     Compute $r = \dfrac{p(\theta^*)q(\theta_{t-1}|\theta^*)}{p(\theta_{t-1})q(\theta^*|\theta_{t-1})}$.
5:     **if** $\mathcal{U}(0, 1) < r$ **then**
6:         Simulate $\mathbf{z}^*$ from model given $\theta^*$.
7:         **if** $\mathbb{1}\{\rho(\mathbf{z}^*, \mathbf{y}) \leq \epsilon\}$ **then**
8:             Set $\theta_t = \theta^*$ else, set $\theta_t = \theta_{t-1}$.
9:         **end if**
10:     **else**
11:         $\theta_t = \theta_{t-1}$.
12:     **end if**
13: **end for**

of an early rejection step [49] can improve the efficiency of the method, since data is only simulated under the model when necessary, as shown in algorithm 5. MCMC ABC is motivated by a desire to keep proposals for $\boldsymbol{\theta}$ in highly probable regions of the posterior. MCMC ABC aims to combat the heavy storage requirements of the ABC rejection sampler, while allowing efficient exploration of the parameter space. The approach requires tuning both the $\epsilon$ parameter and the proposal distribution $q$ for efficient use.

## A.2.3. SMC ABC

In order to improve the low acceptance rate in the basic ABC algorithm, an SMC ABC algorithm was proposed in Sisson *et al.* [14], based on the SMC sampler methodology developed by Del Moral *et al.* [27] (algorithm 6).

---

**Algorithm 6.** SMC ABC algorithm [14,16,27].

---

1: Initialize $\epsilon_t \geq 0$ for $t = 1,....,T$ where $\epsilon_t > \epsilon_{t+1} > 0$.

2: **for** $t = 0$ *to* $T$ **do**

3: for $i = 1$ *to* $N$ **do**

4: if $t = 0$ **then**

5: Sample $\theta^{**}$ from $p(\theta)$.

6: **else**

7: Sample $\theta^*$ from the previous population $\theta_{t-1}^{(i)}$ with normalized weights

 $w_{t-1}^{(i)}$ and use a perturbation kernel $K_t$ to sample $\theta^{**} \sim K_t(\cdot \,|\theta^*)$.

8: **end if**

9: **if** $p(\theta^{**}) = 0$ **then**

10: Go to line 4.

11: **else**

12: Simulate $\mathbf{z}^*$ from model given $\theta^{**}$.

13: **end if**

14: **if** $\rho(\mathbf{z}^*, \mathbf{y}) \geq \epsilon_t$ **then**

15: Go to line 4.

16: **else**

17: Set $\theta_t^{(i)} = \theta^{**}$ and calculate the weight for the particle $\theta_t^{(i)}$:

$$w_t^{(i)} = \begin{cases} 1 & \text{if } t = 0 \\ \dfrac{p(\boldsymbol{\theta}_t^{(i)})}{\sum_{j=1}^{N} w_{t-1}^{(j)} K_t(\boldsymbol{\theta}_t^{(i)} | \boldsymbol{\theta}_{t-1}^{(j)})} & \text{if } t > 0. \end{cases}$$

18: **end if**

19: **end for**

20: set $\epsilon_{t+1}$ to be $\alpha$-quantile of saved distances vector

21: $\epsilon_t = \epsilon_{t+1}$

22: Normalize the weights.

23: **end for**

24: Return particles $\theta_T^{(i)}$.

---

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
