## [Reviewer comments · Royal Society Open Science]

Review History

RSOS-191315.R0 (Original submission)

Review form: Reviewer 1

Is the manuscript scientifically sound in its present form?

No

Are the interpretations and conclusions justified by the results?

Yes

Is the language acceptable?

Yes

Do you have any ethical concerns with this paper?

No

Have you any concerns about statistical analyses in this paper?

Yes

Recommendation?

Major revision is needed (please make suggestions in comments)

Comments to the Author(s)

See attached file (Appendix A).

Review form: Reviewer 2**Is the manuscript scientifically sound in its present form?**

Yes

Are the interpretations and conclusions justified by the results?

Yes

Is the language acceptable?

Yes

Do you have any ethical concerns with this paper?

No

Have you any concerns about statistical analyses in this paper?

No

Recommendation?

Accept with minor revision (please list in comments)

Comments to the Author(s)

Modifications and shortening of the paper have improved the newly submitted version.

I have still some questioning points:

- the authors have implemented the method described by Vaart et al (2018), which seems promising, even though time consuming. The authors should not refer to these results only in the discussion section but should instead show and describe them in the test problem section.
- On this section (3.1), why is MCMC ABC not implemented, as in 3.2 ? This section should be modified accordingly.

Few typos:

-P3 L40: the sentence has no verb

-I do not find Equation(20) (P10 L24)

Review form: Reviewer 3 (Richard Wilkinson)**Is the manuscript scientifically sound in its present form?**

Yes

Are the interpretations and conclusions justified by the results?

Yes

Is the language acceptable?

Yes

Do you have any ethical concerns with this paper?

No

Have you any concerns about statistical analyses in this paper?

No

Recommendation?

Accept as is

Comments to the Author(s)

The authors have done a good job of revising the manuscript. I happy to recommend publication.

Decision letter (RSOS-191315.R0)

16-Dec-2019

Dear Mrs Alahmadi

On behalf of the Editors, I am pleased to inform you that your Manuscript RSOS-191315 entitled "A comparison of approximate versus exact techniques for Bayesian parameter inference in non-linear ODE models" has been accepted for publication in Royal Society Open Science subject to minor revision in accordance with the referee suggestions. Please find the referees' comments at the end of this email.

The reviewers and handling editors have recommended publication, but also suggest some minor revisions to your manuscript. Therefore, I invite you to respond to the comments and revise your manuscript.

- Ethics statement

- Data accessibility

<http://datadryad.org/submit?journalID=RSOS&manu=RSOS-191315>

- Competing interests

- Authors' contributions

- Acknowledgements

- Funding statement

Because the schedule for publication is very tight, it is a condition of publication that you submit the revised version of your manuscript before 25-Dec-2019. Please note that the revision deadline will expire at 00.00am on this date. If you do not think you will be able to meet this date please let me know immediately.

- 1) A text file of the manuscript (tex, txt, rtf, docx or doc), references, tables (including captions) and figure captions. Do not upload a PDF as your "Main Document";

- 2) A separate electronic file of each figure (EPS or print-quality PDF preferred (either format should be produced directly from original creation package), or original software format);
- 3) Included a 100 word media summary of your paper when requested at submission. Please ensure you have entered correct contact details (email, institution and telephone) in your user account;
- 4) Included the raw data to support the claims made in your paper. You can either include your data as electronic supplementary material or upload to a repository and include the relevant doi within your manuscript. Make sure it is clear in your data accessibility statement how the data can be accessed;
- 5) All supplementary materials accompanying an accepted article will be treated as in their final form. Note that the Royal Society will neither edit nor typeset supplementary material and it will be hosted as provided. Please ensure that the supplementary material includes the paper details where possible (authors, article title, journal name).

If your manuscript is newly submitted and subsequently accepted for publication, you will be asked to pay the article processing charge, unless you request a waiver and this is approved by Royal Society Publishing. You can find out more about the charges at <https://royalsocietypublishing.org/rsos/charges>. Should you have any queries, please contact openscience@royalsociety.org.

on behalf of Professor Len Thomas (Associate Editor) and Mark Chaplain (Subject Editor)
openscience@royalsociety.org

Associate Editor Comments to Author (Professor Len Thomas):

The manuscript has been reviewed by three researchers, all of whom were positive about it. However, the first reviewer in particular raised a number of points that need addressing before the manuscript can be published, so I am recommending one more round of revisions. Please address the concerns of this reviewer, and the minor points raised by the second reviewer; I will be happy to look over a revised version and make a recommendation without any further need for peer review. The first reviewer also noted that they obtained an error message when they tried to run the code associated with the paper, so please also double check the code.

Associate Editor: 2
Comments to the Author:
(There are no comments.)

Reviewer comments to Author:
Reviewer: 1

Comments to the Author(s)
See attached file

Reviewer: 2

Comments to the Author(s)
Modifications and shortening of the paper have improved the newly submitted version. I have still some questioning points:
- the authors have implemented the method described by Vaart et al (2018), which seems promising, even though time consuming. The authors should not refer to these results only in the discussion section but should instead show and describe them in the test problem section.
-On this section (3.1), why is MCMC ABC not implemented, as in 3.2 ? This section should be modified accordingly.

Few typos:
-P3 L40: the sentence has no verb
-I do not find Equation(20) (P10 L24)

Reviewer: 3

Comments to the Author(s)
The authors have done a good job of revising the manuscript. I happy to recommend publication.

Author's Response to Decision Letter for (RSOS-191315.R0)

See Appendix B.

Decision letter (RSOS-191315.R1)

27-Jan-2020

Dear Mrs Alahmadi,

It is a pleasure to accept your manuscript entitled "A comparison of approximate versus exact techniques for Bayesian parameter inference in non-linear ODE models" in its current form for publication in Royal Society Open Science. The comments of the reviewer(s) who reviewed your manuscript are included at the foot of this letter.

We note that 'davis.cochrane@monash.edu' is not currently accepting messages from the journal's

emails - please can you ensure the editorial office is provided with a correct and active email address for Dr Cochrane as soon as possible?

Additionally, the Editors have recommended that you update the acknowledgement section of your manuscript to better reflect the support of a number of additional referees (and, indeed, Editors) in the review of your paper - can we suggest that you respond to this with an updated acknowledgement, please?

on behalf of Professor Len Thomas (Associate Editor) and Mark Chaplain (Subject Editor)
openscience@royalsociety.org

Associate Editor Comments to Author (Professor Len Thomas):

Thank-you for responding to the reviews in such a complete way. I am recommending acceptance.

One minor point, however. By my count, the paper has now received 7 reviews (4 from Interface and 3 from Open Science), and yet you only thank two reviewers in your acknowledgements. "We thank Prof. Richard Wilkinson and the anonymous reviewer, whose comments helped us improve our work." Can you please communicate with the editorial team to amend this to a more appropriate statement of the input you received.

Appendix A

Review of “A comparison of approximate versus exact techniques for Bayesian parameter inference in non-linear ODE models”

October 2019

I think the paper makes an interesting and novel point about using ABC for ODE models. Also in my opinion this paper is well suited to Royal Society Open Science, and some of my main criticisms of the previous version don't apply to publication in this venue. However there are still many technical and presentational issues with the paper, so I recommend major corrections.

1 Main issues

1. The paper gives a good criticism of the “ABC with no noise on simulations” approach to inference. Page 4 motivates this with two references: Toni et al (2009) and Gupta et al (2018), which I think is enough for publication in this journal. The other references listed use the “ABC with known simulation noise” setting, which is not investigated by this paper as far as I can tell. I suggest removing these references, or explaining why this is relevant in more detail somewhere in the paper.
2. I don't understand why the authors investigate adaptive SMC ABC algorithms. These

target the same ABC posterior as non-adaptive algorithms, and it seems to me that the problem is with the ABC posterior not the method used to sample from it. (The van der Vaart et al paper does seem more relevant here.) The authors suggest that using a specialised stopping rule with the Prangle (2017) method might approximate the posterior, but it seems simpler to just include noise in the model!

3. Surely some conditions are needed for the highlighted block at the bottom of page 11 to hold? For instance this conclusion would seem not to be the case if some parameters were non-identifiable.
4. The data in Figure 2 seems very unlikely under the normal noise model stated. Firstly, the vast majority of the observations are above the true infection curve. Secondly, no observations appear to be negative.
5. MCMC only converges to its target asymptotically, so statements along the lines of “MCMC required 12401 steps to reach convergence” are incorrect.
6. The tuning choices used for MCMC and MCMC ABC should be summarised (e.g. choice of proposal distributions).
7. The supplied code generally looks excellent, but it did eventually crash when I ran `Run_file_final.R` (The error was `Error in paste0("Runs/ABC_SMC.run_", arg, ".csv") : object 'arg' not found.`)

2 Minor issues

1. Pg 5 “MCMC and ABC... involves sampling the posterior density”. ABC only samples from an approximation to the posterior density.
2. Pg 6 “We then discuss application of these Bayesian frameworks...” This doesn’t seem to follow on logically from the previous sentence. Maybe say something like “In

this section we discuss...”

3. In Equation (2), I don't think the notation $f(\dots)$ has been defined yet.
4. Pg 6 uses the notation α for the quantile used to update ϵ , while in Appendix A it is q instead.
5. “A second departure from standard ABC practice is... Toni directly computes a distance between simulated and observed data, originally using Euclidean distance.” Why is this a departure from standard ABC practice? This seems quite common to me!
6. Pg 13 What are the units of t ?
7. Pg 14 I'm not convinced that MAE is a sensible comparison to use. This would seem to favour over-concentrated posterior approximations.
8. Pg 15 In Table 1, why is the number of SMC ABC iterations “NA”?
9. Pg 15 What is a “challenge tolerance” and why was this particular value used?
10. Pg 21 “ σ^2 is the noise associated with the data”. What is the error model – normal noise?
11. Pg 22 Why is it necessary to transform the parameters to be supported over the real line?
12. Pg 23 “find all distances between these solutions and the true data”. What distance function was used?
13. Pg 23 “we applied SMC ABC for 6 populations with a sequence of tolerance $\epsilon = \dots$ ”. Why this particular sequence? What tolerance was used for MCMC ABC?
14. Pg 24 Table 5's caption mentions mean absolute error but this isn't included in the table.

15. Pg 24 “Estimation of the noise parameter is standard using exact Bayesian inference (MCMC), but not with the highly popular ABC based approaches we have investigated here”. In my experience noise parameter estimation is standard in applications of ABC, and papers which avoid it are unusual.
16. Pg 25 “We can see in all examples presented in this paper that the computational time consumed by MCMC ABC is shorter compared to the other methods... This is a significant advantage...” I think MCMC ABC was only used once in the paper (Tables 4 and 5), which is not enough to make a conclusion like this.
17. Pg 26 “the problem with this algorithm is that there is no existing criteria to identify an appropriate iteration at which to terminate”. I think this sentence is a bit misleading. The authors have proposed an unusual application of this algorithm to the case where no noise is added to ABC simulations. I would argue that the lack an appropriate termination criterion in this case is a problem with this unusual application, not an general problem of the algorithm.
18. Pg 26-27 The extra experiments involving Figure 10 should be in a results section, not the conclusion.
19. Pg 26 “The comparison conducted in this paper demonstrates that using exact Bayesian inference (MCMC) for ODE parameter estimation is a practical alternative”. This is very well known and not a novel finding. See for example Gelman, Bois and Jiang (1996) “Physiological pharamacokinetic analysis using population modeling and informative prior distributions”.
20. Pg 28 “Both MCMC and SMC ABC method incur similar computational cost” MCMC took 6.6 minutes while SMC ABC took 11.2 minutes - roughly double the cost.
21. Pg 27 “the needs to solve the ODE models too many times”. “Too many” compared to what?

22. Pg 27 “in the second example we found that more effort is needed to construct the likelihood functions when applying MCMC” I don’t understand this comment – isn’t the likelihood just based on using normal noise again? This doesn’t seem like much effort.

3 Possible typos

1. Pg 3 “An SMC ABC approach developed by” should be “An SMC ABC approach was developed by”
2. Pg 8 What’s \hat{z}_n ?
3. Pg 12 “The ABC and MCMC techniques described in Section 2...” Section 2 didn’t describe any MCMC techniques!
4. Pg 13 “The parameter of interest is” should be “The parameters of interest are”
5. Pg 15 “SMC ABC consumed the longest run times amongst the methods”, “. . . has the smallest variance compared with other methods”. These sentences should be reworded to reflect that only two methods were compared.
6. Pg 20 “no-prior” should be “no prior”?

Appendix B

Responses to referee reports - manuscript RSOS-191315

Amani A. Alahmadi, Jennifer A. Flegg, Davis G. Cochrane,
Christopher C. Drovandi and Jonathan M. Keith

December 25, 2019

We thank the reviewers for their helpful comments. Below we systematically address the comments and highlight changes made to the resubmitted manuscript. Page references made in our response are those from the resubmitted document. Changes to the manuscript are highlighted in yellow.

Referee # 1

General comments

1. I think the paper makes an interesting and novel point about using ABC for ODE models. Also in my opinion this paper is well suited to Royal Society Open Science.

We would like to thank the reviewer for this positive evaluation.

Main issues

1. The paper gives a good criticism of the "ABC with no noise on simulations" approach to inference. Page 4 motivates this with two references: Toni et al (2009) and Gupta et al (2018), which I think is enough for publication in this journal. The other references listed use the "ABC with known simulation noise" setting, which is not investigated by this paper as far as I can tell. I suggest removing these references, or explaining why this is relevant in more detail somewhere in the paper.

We removed Silk et al. (2013); da Costa et al. (2018); Costa et al. (2018) as suggested because they assumed known noise and need more investigation, but we kept the others (Barnes et al. (2011); Toni and Stumpf (2009); Sun et al. (2015)) as they do not give details regarding their assumptions about the noise.

2. I do not understand why the authors investigate adaptive SMC ABC algorithms. These target the same ABC posterior as non-adaptive algorithms, and it seems to me that the problem is with the ABC posterior not the method used to sample from it. (The Vaart et al. (2018) paper does seem more relevant here.) The authors suggest that using a specialised stopping rule with the Prangle (2017) method might approximate the posterior, but it seems simpler to just include noise in the model!

We agree that also non-adaptive algorithms will not accurately reflect the uncertainties in parameter values, but we used different adaptive ABC methods such as SMC ABC and MCMC ABC, to confirm that even using the most recent ABC methods fail to accurately approximate the posterior distributions. Regarding to the stopping rule we mention it among others methods that been used in the literature to

terminate algorithm and we "found that none of these methods terminate the algorithm (in ODE case) in such a way as to produce the correct shape and spread of the posterior distribution" as we stated on page 17 and 18.

3. Surely some conditions are needed for the highlighted block at the bottom of page 11 to hold? For instance this conclusion would seem not to be the case if some parameters were non-identifiable.

In response to this comment we clarify in the paper that under an ideal conditions the highlighted block will be hold. In addition, it is standard in SMC ABC that extra reductions of the tolerance ϵ_0 will leads to low acceptance rates without adding a significant improvement to the ABC posterior.

4. The data in Figure 2 seems very unlikely under the normal noise model stated. Firstly, the vast majority of the observations are above the true infection curve. Secondly, no observations appear to be negative.

We randomly added normal noise to the true ODE solution and because the data is represents the proportion of infected individuals negative value (less than zero) is not appropriate. Therefore, we repeat the generation of the noise randomly until we have positive data.

5. MCMC only converges to its target asymptotically, so statements along the lines of "MCMC required 12401 steps to reach convergence" are incorrect.

Fixed, page 15.

6. The tuning choices used for MCMC and MCMC ABC should be summarised (e.g. choice of proposal distributions).

Fixed, page 14 and 23.

7. The supplied code generally looks excellent, but it did eventually crash when I ran Run file final.R.

Fixed.

Minor issues

1. Pg 5 "MCMC and ABC. . . involves sampling the posterior density". ABC only samples from an approximation to the posterior density.

Fixed, page 5.

2. Pg 6 "We then discuss application of these Bayesian frameworks. . . ." This does not seem to follow on logically from the previous sentence. Maybe say something like "In this section we discuss...?"

Fixed, page 5.

3. In Equation (2), I do not think the notation $f(\cdot, \cdot)$ has been defined yet.

Fixed, page 6.

4. Pg 6 uses the notation α for the quantile used to update, while in Appendix A it is q instead.

Fixed.

5. "A second departure from standard ABC practice is. . . Toni directly computes a distance between simulated and observed data, originally using Euclidean distance."? Why is this a departure from standard ABC practice? This seems quite common to me!

Our claim here is that the common practice with ABC approach is using a discrepancy function based on the distance between vectors of summary statistics $s(\mathbf{z}^)$ and $s(\mathbf{y})$ not directly computes a distance between simulated and observed data as in Toni et al. (2009) approach. Because the summary statistics have much lower dimension than the simulated and observed data vectors \mathbf{z}^* and \mathbf{y} . This explanation is appear on paragraph 3 page 11.*

6. Pg 13 What are the units of t?

The units of the time is (weeks), we added this on page 13.

7. Pg 14 I'm not convinced that MAE is a sensible comparison to use. This would seem to favour over-concentrated posterior approximations.

We used range of performance measures such as CPU times, the number of iterations and the median of the posterior in addition to the mean absolute value, which give a good comparisons of the methods. In response to this comment, we agree with the reviewer and we clarify in the paper that MAE may favour over-concentrated posterior approximations.

8. Pg 15 In Table 1, why is the number of SMC ABC iterations "NA"?

Fixed, we add the number of iterations.

9. Pg 15 What is a "challenge tolerance" and why was this particular value used?

In example 1 the challenge tolerance been chosen by finding the distance between the true ODE solution and the generate observations y . We clarify this at page 15.

10. Pg 21 σ is the noise associated with the data?. What is the error model ? normal noise?

True, we assumed normal noise and we clarify this at page 22.

11. Pg 22 Why is it necessary to transform the parameters to be supported over the real line?

The parameters were transformed to improve the acceptance rate of the proposals, we clarify this at page 23.

12. Pg 23 "find all distances between these solutions and the true data". What distance function was used?

Fixed, page 24.

13. Pg 23 "we applied SMC ABC for 6 populations with a sequence of tolerance" Why this particular sequence? What tolerance was used for MCMC ABC?

The sequence of tolerance have been chosen adaptively, we clarify this on page 24. Regarding the way to choose MCMC ABC tolerance was demonstrated on the last paragraph on page 24.

14. Pg 24 Table 5 caption mentions mean absolute error but this is not included in the table.

Fixed.

15. Pg 24 "Estimation of the noise parameter is standard using exact Bayesian inference (MCMC), but not with the highly popular ABC based approaches we have investigated here". In my experience noise parameter estimation is standard in applications of ABC, and papers which avoid it are unusual.

In response to this comments we have revised the sentence as follows on page 25: "Estimation of the noise parameter is standard using exact Bayesian inference (MCMC), but not with the current practice with ABC based approaches when applying to a system of ODEs that we investigated here". This clarify that the neglecting of the estimation of the noise when using ABC based approach have been done in some literatures when it applied on ODEs system.

16. Pg 25 "We can see in all examples presented in this paper that the computational time consumed by MCMC ABC is shorter compared to the other methods. . . This is a significant advantage. . . " I think MCMC ABC was only used once in the paper (Tables 4 and 5), which is not enough to make a conclusion like this.

We agree with the reviewer and in response to this comment we remove this sentence.

17. Pg 26 "the problem with this algorithm is that there is no existing criteria to identify an appropriate iteration at which to terminate". I think this sentence is a bit misleading. The authors have proposed an unusual application of this algorithm to the case where no noise is added to ABC simulations. I would argue that the lack an appropriate termination criterion in this case is a problem with this unusual application, not an general problem of the algorithm.

To the best of our knowledge there is no current SMC ABC approach applied to the ODE models that terminates the iteration before it is shrink to point estimate as we explained in the paper. Estimating the number of population T required in such way the approximated ABC posterior reflects the noise on the data can be investigated further in a future work.

18. Pg 26-27 The extra experiments involving Figure 10 should be in a results section, not the conclusion.

Fixed as suggested.

19. Pg 26 "The comparison conducted in this paper demonstrates that using exact Bayesian inference (MCMC) for ODE parameter estimation is a practical alternative". This is very well known and not a novel finding. See for example Gelman et al. (1996).

We agree with the reviewer that the study conducted by Gelman et al. (1996) have observed that using MCMC with complex model has many features, but in our paper we concluded to this finding after the comparison between the exact and approximate Bayesian inference when we dealing with a system of ODEs. One aim is to shed light on which method is perform better. In response we added the provided reference on page 27.

20. Pg 28 "Both MCMC and SMC ABC method incur similar computational cost? MCMC took 6.6 minutes while SMC ABC took 11.2 minutes - roughly double the cost.

Fixed, page 28.

21. Pg 27 "the needs to solve the ODE models too many times". Too many? compared to what?

Fixed, page 28.

22. Pg 27 "in the second example we found that more effort is needed to construct the likelihood functions when applying MCMC? I do not understand this comment ? is not the likelihood just based on using

normal noise again? This does not seem like much effort.

That's true for example 1, but for example 2 the likelihood was not straightforward because the ODEs system is coupled and complex. While computation of the Gaussian terms is straightforward, solving the system of equations to obtain the means is not.

Possible typos

1. Pg 3 "An SMC ABC approach developed by" should be "An SMC ABC approach was developed by".
Fixed as suggested.
2. Pg 8 What is " z_n ".
 z_n is the n - th simulated data as we defined it on page 7.
3. Pg 12 "The ABC and MCMC techniques described in Section 2...? Section 2 did not describe any MCMC techniques!.
Fixed, page 12.
4. Pg 13 "The parameter of interest is" should be "The parameters of interest are".
Fixed, page 13.
5. Pg 15 "SMC ABC consumed the longest run times amongst the methods", ". . . has the smallest variance compared with other methods?. These sentences should be reworded to reflect that only two methods were compared.
Fixed.
6. Pg 20 "no-prior" should be "no prior"?
Fixed.

Referee # 2

General comments

" Modifications and shortening of the paper have improved the newly submitted version".

We would like to thank the reviewer for this positive evaluation.

Minor remarks

1. The authors have implemented the method described by Vaart et al. (2018), which seems promising, even though time consuming. The authors should not refer to these results only in the discussion section but should instead show and describe them in the test problem section.

As mentioned in response to Referee # 1, we have moved the implementation of Vaart et al. (2018) method to the results section.

2. On this section (3.1), why is MCMC ABC not implemented, as in 3.2 ? This section should be modified accordingly.

In the initial manuscript we implemented MCMC ABC on the example in section (3.1), but we removed it as suggested with initial referees to reduce the length of the paper.

3. P3 L40: the sentence has no verb.

Fixed.

4. I do not find Equation(20) (P10 L24).

Fixed.

Referee # 3

General comments

”The authors have done a good job of revising the manuscript. I happy to recommend publication.”

We would like to thank the reviewer for this positive recommendation.

References

- Barnes, C. P., Silk, D., and Stumpf, M. P. (2011). Bayesian design strategies for synthetic biology. *Interface focus*, 1(6):895–908. 1
- Costa, J. M., Orlande, H. R., Lione, V. O., Lima, A. G., Cardoso, T. C., and Varon, L. A. (2018). Simultaneous model selection and model calibration for the proliferation of tumor and normal cells during in vitro chemotherapy experiments. *Journal of Computational Biology*, 25(12):1285–1300. 1
- da Costa, J. M. J., Orlande, H. R. B., and da Silva, W. B. (2018). Model selection and parameter estimation in tumor growth models using approximate bayesian computation-abc. *Computational and Applied Mathematics*, 37(3):2795–2815. 1
- Gelman, A., Bois, F., and Jiang, J. (1996). Physiological pharmacokinetic analysis using population modeling and informative prior distributions. *Journal of the American Statistical Association*, 91(436):1400–1412. 4
- Silk, D., Filippi, S., and Stumpf, M. P. (2013). Optimizing threshold-schedules for sequential approximate bayesian computation: applications to molecular systems. *Statistical applications in genetics and molecular biology*, 12(5):603–618. 1
- Sun, L., Lee, C., and Hoeting, J. A. (2015). Parameter inference and model selection in deterministic and stochastic dynamical models via approximate bayesian computation: modeling a wildlife epidemic. *Environmetrics*, 26(7):451–462. 1
- Toni, T. and Stumpf, M. P. (2009). Simulation-based model selection for dynamical systems in systems and population biology. *Bioinformatics*, 26(1):104–110. 1
- Toni, T., Welch, D., Strelkowa, N., Ipsen, A., and Stumpf, M. P. (2009). Approximate Bayesian computation scheme for parameter inference and model selection in dynamical systems. *Journal of The Royal Society Interface*, 6(31):187202. 3
- Vaart, E., Prangle, D., and Sibly, R. M. (2018). Taking error into account when fitting models using approximate bayesian computation. *Ecological Applications*, 28(2):267–274. 1, 5